# L-CUBE: Isolating Long-Context Capacity from Knowledge with Controllable Mutual Information Scaling

**Zhuo Chen** [* 1 2 3]  **Oriol Mayné i Comas** [* 2 4 5]  **Zhuotao Jin** [6 7]  **Di Luo** [1 2 8 9]  **Marin Soljačić** [1 2 10]

## Abstract

Evaluating long-context language models on natural language conflates architectural capacity to capture dependencies with semantic knowledge and vocabulary statistics. When models fail at long contexts, we cannot determine whether failures stem from fundamental architectural limitations or insufficient domain knowledge, preventing clean diagnosis of efficient architectures before expensive training on real data. We introduce **L-CUBE** (LONG-CONTEXT UTILIZATION BENCHMARK), a synthetic benchmark that isolates dependency-capturing capacity from semantic knowledge through hierarchical Gaussian sequences with controllable bipartite mutual information scaling. The generator provides exact ground-truth conditionals that scale efficiently to arbitrarily long sequences, enabling unconfounded evaluation via conditional KL divergence rather than perplexity alone. We define *long-context utilization* to measure the amount of available predictive information that models extract as context grows. Experiments across transformers, state space models, and efficient alternatives validate $L^2M$ capacity theory predictions and uncover new phenomena. L-CUBE enables practitioners to test whether a particular design will maintain long-context capability at target sequence lengths before committing to real-data training.

---

[*]Equal contribution  [1]NSF AI Institute for Artificial Intelligence and Fundamental Interactions [2]Department of Physics, MIT [3]Institute for Data, Systems and Society, MIT [4]Department of Computer Science, Polytechnic University of Catalonia [5]Department of Physics, Polytechnic University of Catalonia [6]Harvard School of Engineering and Applied Sciences [7]Department of Materials Science and Engineering, MIT [8]Department of Physics, Harvard University [9]Department of Electrical and Computer Engineering, UCLA [10]Research Laboratory of Electronics, MIT. Correspondence to: Zhuo Chen <chenzhuo@mit.edu>.

*Proceedings of the 43rd International Conference on Machine Learning*, Seoul, South Korea. PMLR 306, 2026. Copyright 2026 by the author(s).

## 1. Introduction

Long-context language models are typically evaluated on natural-language benchmarks where improvements in next-token prediction conflate multiple effects: a model's architectural capacity to capture long-range dependencies, its semantic knowledge of the domain, and the quality of its learned token statistics. When a model fails to effectively utilize long contexts, it is unclear whether the failure stems from fundamental architectural limitations or insufficient training on domain-specific knowledge (Fig. 1). This ambiguity makes it difficult to diagnose architectural weaknesses, guide design choices, or predict whether a given architecture will scale to longer contexts.

This evaluation challenge has practical urgency. Transformers with full attention (Vaswani et al., 2017) successfully model long contexts but scale poorly in memory and computation, making them prohibitively expensive beyond moderate sequence lengths. State space models (SSMs) (Gu & Dao, 2024; Dao & Gu, 2024; Peng et al., 2023), sparse attention mechanisms (Beltagy et al., 2020; Zaheer et al., 2020), and other efficient alternatives promise linear or subquadratic scaling. Existing long-context benchmarks such as SCROLLS (Shaham et al., 2022), LongBench (Bai et al., 2024), and needle-in-haystack style evaluations (Liu et al., 2024) evaluate end-to-end performance but cannot isolate architectural capacity from knowledge: failures may stem from missing domain knowledge rather than structural limits (see Appx. A for additional discussion of related benchmarks and evaluation approaches). Evaluating whether these efficient architectures maintain long-context capability currently requires expensive training on real data. Without controlled evaluation that separates dependency structure from semantics, practitioners face high costs and uncertainty when selecting architectures for long-context applications.

Recent theoretical work, the Long-context Language Modeling ($L^2M$) theory (Chen et al., 2025), provides predictions about when architectures should fail at long contexts based on their history state capacity. The theory proves a necessary condition: if bipartite mutual information $I^{\mathrm{BP}}$ between a prefix and suffix grows with sequence length, then the model's effective history state dimension must scale correspondingly for the model to capture this long-range depen-

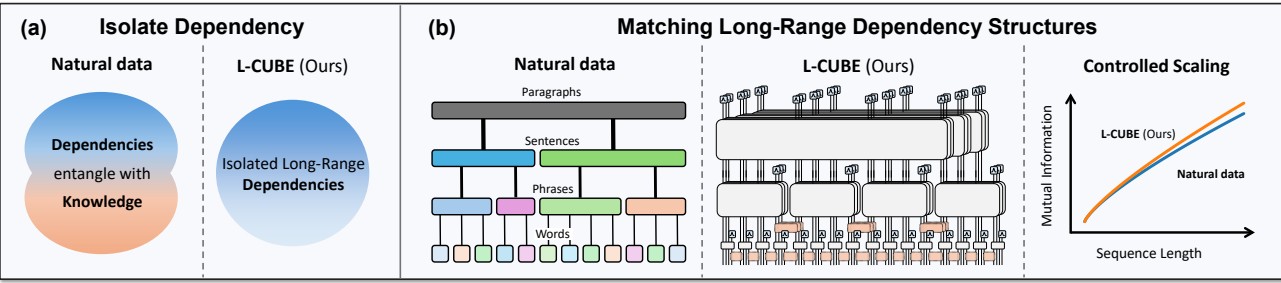

*Figure 1.* (a) Natural language entangles semantic knowledge with architectural requirements for capturing long-range dependencies. When models fail on natural text, we cannot diagnose whether failures stem from insufficient knowledge or architectural capacity limits. L-CUBE isolates long-range dependencies by construction, removing semantic confounders. (b) Both natural language and L-CUBE exhibit hierarchical compositional structure. L-CUBE explicitly constructs this hierarchy through a multi-scale generator that produces controllable bipartite mutual information scaling matching natural language power-law growth.

dence. However, the $L^2M$ condition is necessary but not sufficient: even architectures that theoretically satisfy the history state capacity requirement must be empirically tested to confirm they effectively capture long-range dependencies in practice. Natural language lacks ground-truth conditional distributions and controllable information scaling, making it impossible to cleanly test architectural capacity separate from confounding factors such as semantic knowledge and vocabulary statistics.

We introduce L-CUBE (LONG-CONTEXT UTILIZATION BENCHMARK)[1], a synthetic benchmark designed to isolate and measure how well language model architectures capture hierarchical long-range dependencies. L-CUBE uses continuous Gaussian sequences to eliminate vocabulary knowledge and domain semantics as confounds, provides exact conditional distributions $p(w_t \mid w_{1:t-1})$ enabling direct computation of conditional KL divergence, and constructs sequences through a multi-scale hierarchical process mirroring natural language's compositional structure. The benchmark is parameterized to control bipartite mutual information $I^{BP}$ as a function of sequence length; in the default configuration, we match the sub-volume power-law growth ($I^{BP} \sim L^{0.7\text{-}0.8}$ from Chen et al. (2025)) observed in natural language. In the default sub-volume configuration, the construction requires $O(L)$ time and space, enabling practical evaluation at sequence lengths from hundreds to tens of thousands of tokens.

We define *long-context utilization* $U(\boldsymbol{X}; \boldsymbol{Y})$ to measure how much predictive information a model extracts from context $\boldsymbol{X}$ when predicting target $\boldsymbol{Y}$, compared against the *available gain* $G(\boldsymbol{X}; \boldsymbol{Y}) = I(\boldsymbol{X}; \boldsymbol{Y})$. The difference forms a *KL residual* that quantifies how architectural limitations prevent models from utilizing available context.

We evaluate representative architectures across transformers with full attention (GPT-2, GPT-NeoX, Qwen3), state

space models (Mamba, Mamba2, RWKV), log-linear attention (LLA), and windowed attention (Mistral), spanning sequence lengths from $256$ to $16{,}384$ tokens and model sizes from $64M$ to $1.4B$ parameters. The benchmark cleanly isolates architectural limitations predicted by $L^2M$ theory. Sufficiently large transformers track the ground-truth bipartite mutual information across all tested lengths, while state space models plateau and decline as sequence length increases, exhibiting the failure mode predicted by fixed history state capacity. Beyond validating theory, L-CUBE reveals that when viewed through computational complexity, different architectural families exhibit surprisingly similar scaling in how utilized context length grows with compute. Most importantly, L-CUBE enables practitioners to test whether a particular architecture will maintain long-context capability at target sequence lengths before committing to expensive training on real data.

## 2. Preliminaries

We use standard information-theoretic notation. Random variables are denoted by uppercase letters and realizations by lowercase. For a length-$L$ sequence $W_{1:L} := (W_1, \ldots, W_L)$, we use subsequence notation $W_{i:j} := (W_i, \ldots, W_j)$ and boldface to denote blocks when indices are clear from context (e.g., $\boldsymbol{W} := W_{1:L}$). We write $D_{KL}(\cdot\|\cdot)$ for KL divergence, $I(\cdot; \cdot)$ for mutual information, and $H(\cdot)$ for entropy (or differential entropy for continuous variables). When it matters, we use the subscript convention $H_p(\cdot)$ for entropy under distribution $p$, and the conditional cross-entropy notation

$$H(p_{\boldsymbol{Y}|\boldsymbol{X}}, q_{\boldsymbol{Y}|\boldsymbol{X}}) := -\mathbb{E}_{p_{\boldsymbol{XY}}} \log q(\boldsymbol{Y} \mid \boldsymbol{X}). \quad (1)$$

### 2.1. Bipartite mutual information

A central object in the Chen et al. (2025) is the *bipartite mutual information*, also called *predictive information*, which measures how much a prefix predicts a suffix. Fix a split

---

[1] https://github.com/LSquaredM/L-CUBE.

location $\ell \in \{1, \ldots, L-1\}$ and define the two adjacent blocks

$$\boldsymbol{X} := W_{1:\ell}, \qquad \boldsymbol{Y} := W_{\ell+1:L}.$$

The bipartite mutual information is

$$I_{\ell;L}^{\mathrm{BP}} := I(\boldsymbol{X}; \boldsymbol{Y}). \tag{2}$$

Applying the standard identity $I(\boldsymbol{X}; \boldsymbol{Y}) = H(\boldsymbol{Y}) - H(\boldsymbol{Y} \mid \boldsymbol{X})$ yields

$$H(\boldsymbol{Y}) = H(\boldsymbol{Y} \mid \boldsymbol{X}) + I_{\ell;L}^{\mathrm{BP}}, \tag{3}$$

so $I_{\ell;L}^{\mathrm{BP}}$ is exactly the reduction in uncertainty about the suffix due to observing the prefix. Prior works (Chen et al., 2025; Dębowski, 2015) report that for natural language, the bipartite mutual information $I_{\ell;n\ell}^{\mathrm{BP}}$ grows with $\ell$ over broad regimes, typically following a sub-volume law $I_{\ell;n\ell}^{\mathrm{BP}} \sim \ell^{\beta}$.

## 2.2. History state and an information capacity bound

Autoregressive models parameterize next-token conditionals through an intermediate *history state* that summarizes the past. Concretely, these models have the form

$$\boldsymbol{z}_{t-1} := \boldsymbol{f}(w_{1:t-2}),$$
$$q(w_t \mid w_{1:t-1}) := q(w_t \mid w_{t-1}, \boldsymbol{z}_{t-1}), \tag{4}$$

where $\boldsymbol{z}_{t-1}$ is the state available immediately before predicting $w_t$. The L$^2$M condition (Chen et al., 2025) proves that the bipartite mutual information representable by such models at a split $\ell$ is upper bounded by the history state $\dim(\boldsymbol{z}_\ell)$ up to a multiplicative constant factor and a small vocabulary-dependent additive constant. Consequently, when the data distribution requires $I_{\ell;n\ell}^{\mathrm{BP}}$ to increase with $\ell$, architectures whose effective state dimension does not scale appropriately with $\ell$ must eventually fail to capture the required long-range dependence at sufficiently large lengths. This motivates a benchmark whose long-range information requirements are known and tunable. We refer to Chen et al. (2025) for formal statements and proofs.

**Remark (continuous variables).** Continuous variables appear throughout this work. Our primary scores are constructed from *differences* of expected log densities across lengths or models and can be expressed in terms of differences of KL terms; they do not depend on absolute values of differential entropy. Technical details are deferred to Appx. B.

## 3. Separating Dependence from Semantics

L-CUBE is designed to study long-context *utilization* rather than semantic knowledge. The question it isolates is: given a sequence distribution with a known amount of long-range predictive information, how much of that information can a model architecture actually *utilize* as context length increases?

### 3.1. Natural language confounders

On natural text, next-token negative log-likelihood (NLL) conflates multiple effects. Some uncertainty is semantic and local, while some is genuinely long-range. For example, failing to predict "Paris" after "The capital of France is" reflects lack of semantic knowledge, whereas failing to resolve pronouns to entities mentioned paragraphs earlier may reflect inability to capture long-range structure. More importantly for diagnosis, the true conditional distribution of language is unknown, so we cannot directly evaluate the conditional mismatch

$$D_{\mathrm{KL}}\left(p_{W_t \mid W_{1:t-1}} \,\|\, q_{W_t \mid W_{1:t-1}}\right). \tag{5}$$

This also obscures the Bayes-optimal achievable NLL at a given context length, making it difficult to tell how well models capture the long-range correlations that are in principle learnable. Furthermore, the intrinsic conditional entropy of natural language decreases with context length as longer contexts reduce next-token uncertainty, so even the baseline prediction difficulty shifts with scale. Changes in NLL with longer contexts can therefore reflect data coverage, optimization, memorization effects, or distributional shifts in addition to, or instead of, architectural limits on long-range dependence.

### 3.2. Removing confounders with L-CUBE

L-CUBE removes these confounders by construction (Fig. 1). It generates sequences from a fully specified stochastic process with no external semantics and provides the ground-truth next-token conditionals of that process. This makes it possible to evaluate, at each position, the mismatch between a model conditional $q(w_t \mid w_{1:t-1})$ and the known conditional $p(w_t \mid w_{1:t-1})$.

The generator is parameterized so that the long-range information content can be controlled via the scaling of $I_{\ell;L}^{\mathrm{BP}}$ with length $L$ and split location $\ell$. In the default configuration, L-CUBE is tuned so that the growth of $I^{\mathrm{BP}}$ (with fixed $\ell/L$) lies in the same qualitative sub-volume regime reported for natural language in Chen et al. (2025). Fig. 2 shows this comparison: the scaling exponent in our default configuration (sub-volume Gaussian) is $\beta = \log(3)/\log(4) \approx 0.79$, matching the range observed in natural language. This yields a controlled proxy setting in which long-range predictive information grows with context length, while the ground-truth conditionals remain available.

### 3.3. Utilization metric

Because $p$ is known in L-CUBE, one can measure conditional KL divergences directly. However, we primarily report a *long-context utilization* score that isolates what the *presence of long-range context* adds to prediction quality. Concretely, we compare a model trained to predict a target

block without any context to a model trained to predict the same target block given an additional context block. This subtraction cancels length-independent effects such as local modeling quality and yields an information-like quantity that we interpret as a *cross mutual information* between context and target under the learned conditionals.

Let $\boldsymbol{X}$ denote a context block and $\boldsymbol{Y}$ the subsequent target block. We will compare prediction of the *same* $\boldsymbol{Y}$ with and without access to $\boldsymbol{X}$.[2] Let $q_y$ denote a model trained on samples of $\boldsymbol{Y}$ alone (no context), and let $q_{xy}$ denote a model trained on paired samples $(\boldsymbol{X}, \boldsymbol{Y})$ to predict $\boldsymbol{Y}$ conditioned on $\boldsymbol{X}$. We evaluate both models under the data-generating process $p$:

$$H_p(\boldsymbol{Y}) := -\mathbb{E}_p[\log p(\boldsymbol{Y})], \qquad (6)$$

$$H_p(\boldsymbol{Y} \mid \boldsymbol{X}) := -\mathbb{E}_p[\log p(\boldsymbol{Y} \mid \boldsymbol{X})], \qquad (7)$$

$$H(p_{\boldsymbol{Y}}, q_{y\boldsymbol{Y}}) := -\mathbb{E}_p[\log q_y(\boldsymbol{Y})], \qquad (8)$$

$$H(p_{\boldsymbol{Y}\mid\boldsymbol{X}}, q_{xy\,\boldsymbol{Y}\mid\boldsymbol{X}}) := -\mathbb{E}_p[\log q_{xy}(\boldsymbol{Y} \mid \boldsymbol{X})], \quad (9)$$

where we implicitly average over the relevant $p(\boldsymbol{X})$ when conditioning is present.

We define the *available* predictive gain from revealing $\boldsymbol{X}$ as

$$G(\boldsymbol{X}; \boldsymbol{Y}) := H_p(\boldsymbol{Y}) - H_p(\boldsymbol{Y} \mid \boldsymbol{X}) = I_p(\boldsymbol{X}; \boldsymbol{Y}), \quad (10)$$

which is the mutual information between $\boldsymbol{X}$ and $\boldsymbol{Y}$ under the data-generating process. Our primary *long-context utilization* score is the *realized* gain captured by the trained models:

$$U(\boldsymbol{X}; \boldsymbol{Y}) := \mathbb{E}_p[\log q_{xy}(\boldsymbol{Y} \mid \boldsymbol{X}) - \log q_y(\boldsymbol{Y})]. \quad (11)$$

Equivalently,

$$U(\boldsymbol{X}; \boldsymbol{Y}) = H(p_{\boldsymbol{Y}}, q_{y\boldsymbol{Y}}) - H(p_{\boldsymbol{Y}\mid\boldsymbol{X}}, q_{xy\,\boldsymbol{Y}\mid\boldsymbol{X}}), \quad (12)$$

so $U(\boldsymbol{X}; \boldsymbol{Y})$ can be viewed as a *cross mutual information*: it has the same formal structure as $I_p(\boldsymbol{X}; \boldsymbol{Y})$, but replaces the true conditionals by those induced by the trained model family. Unlike $I_p(\boldsymbol{X}; \boldsymbol{Y})$, $U(\boldsymbol{X}; \boldsymbol{Y})$ is not constrained to be nonnegative; negative values indicate that conditioning worsens the learned fit relative to the no-context baseline.

To make explicit how model mismatch reduces the realized gain, define the *KL residual*

$$\Delta_{\mathrm{KL}}(\boldsymbol{X}; \boldsymbol{Y}) := D_{\mathrm{KL}}(p_{\boldsymbol{Y}\mid\boldsymbol{X}} \| q_{xy\,\boldsymbol{Y}\mid\boldsymbol{X}}) - D_{\mathrm{KL}}(p_{\boldsymbol{Y}} \| q_{y\boldsymbol{Y}}). \quad (13)$$

Using $H(p, q) = H_p + D_{\mathrm{KL}}(p\|q)$, we obtain the decomposition

$$U(\boldsymbol{X}; \boldsymbol{Y}) = G(\boldsymbol{X}; \boldsymbol{Y}) - \Delta_{\mathrm{KL}}(\boldsymbol{X}; \boldsymbol{Y}). \quad (14)$$

---

[2]Here $\boldsymbol{X}$ denotes a token block and is unrelated to the data distribution $p$.

In particular, if conditioning on $\boldsymbol{X}$ does not adversely affect the fit, so that $\Delta_{\mathrm{KL}}(\boldsymbol{X}; \boldsymbol{Y}) \approx 0$, then $U(\boldsymbol{X}; \boldsymbol{Y}) \approx I_p(\boldsymbol{X}; \boldsymbol{Y})$. Conversely, if the model family fails to exploit $\boldsymbol{X}$, then $U(\boldsymbol{X}; \boldsymbol{Y})$ can remain small even when the available gain $G(\boldsymbol{X}; \boldsymbol{Y})$ is large.

Finally, we define the *relative residual*, which normalizes the KL residual by the available mutual information:

$$\rho(\boldsymbol{X}; \boldsymbol{Y}) := \frac{\Delta_{\mathrm{KL}}(\boldsymbol{X}; \boldsymbol{Y})}{G(\boldsymbol{X}; \boldsymbol{Y})} = \frac{\Delta_{\mathrm{KL}}(\boldsymbol{X}; \boldsymbol{Y})}{I_p(\boldsymbol{X}; \boldsymbol{Y})}. \quad (15)$$

By Eq. (14), this can be written as $\rho(\boldsymbol{X}; \boldsymbol{Y}) = 1 - U(\boldsymbol{X}; \boldsymbol{Y})/G(\boldsymbol{X}; \boldsymbol{Y})$. The relative residual provides a normalized measure of how much of the available information the model fails to capture, and we use it to define the maximum utilized context length in Sec. 5.

## 4. L-CUBE: Construction and Properties

### 4.1. Design principles

Natural language exhibits structure at multiple resolutions: words combine into phrases, phrases into sentences, sentences into paragraphs, and so forth [Fig. 1 (b)]. This hierarchical organization means that long-range dependencies are mediated through compositional structure rather than flat distance.

L-CUBE is a family of fully specified Gaussian sequence distributions constructed to isolate this hierarchical dependence geometry from semantic knowledge while remaining exactly evaluable, partly inspired by the MERA tensor network (Vidal, 2007; 2008) (further explained in Appx. D). Our construction provides explicit control over long-range dependence structure, which we characterize using the bipartite mutual information $I_{\ell;L}^{\mathrm{BP}} := I(W_{1:\ell}; W_{\ell+1:L})$.

The construction targets three goals: (i) tune how the bipartite mutual information $I^{\mathrm{BP}}$ grows with sequence length $L$, allowing for not only matching the sub-volume (power-law) growth patterns observed in natural language but also sweeping across distinct regimes from constant to volume-law (Fig. 2, with additional results in Appx. C); (ii) enable efficient exact evaluation of the ground-truth conditionals $p(w_t \mid w_{1:t-1})$, which provides unconfounded measurement of how well models capture dependencies; and (iii) scale to large $L$ with an implementation that supports efficient, parallelizable generation.

### 4.2. Generative construction

We construct sequences through a hierarchical process mirroring how structure compounds in natural language (Alg. LCUBEGENERATOR). Fix block size $n \geq 2$, reuse channels $1 \leq m \leq n$, depth $D \in \mathbb{N}$, and per-scale linear maps $\{\mathcal{M}^{(d)}\}_{d=1}^D$ with $\mathcal{M}^{(d)} \in \mathbb{R}^{n \times n}$. The generator out-

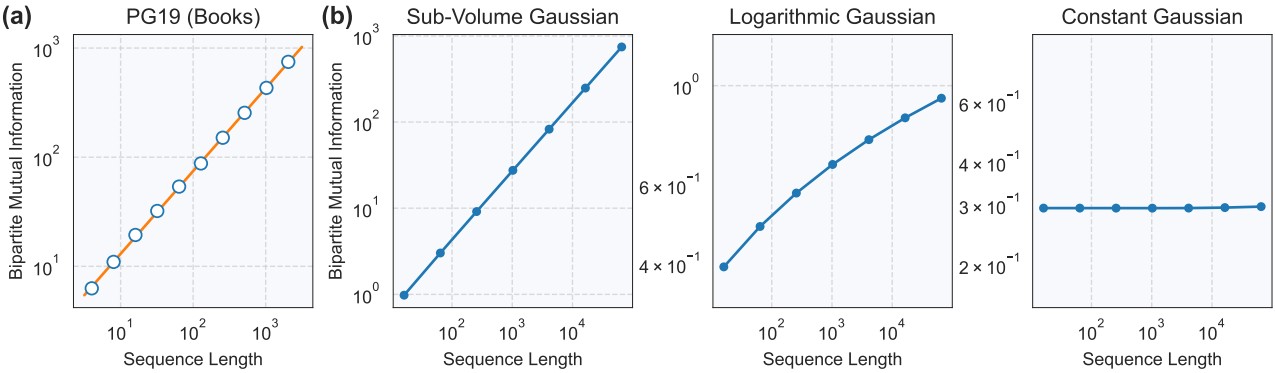

*Figure 2.* (a) Empirical scaling of $I_{L/2;L}^{\mathrm{BP}}$ versus sequence length $L$ for natural language corpora, reproduced from Chen et al. (2025), showing sub-volume power-law growth with exponent $\beta \approx 0.7\text{-}0.8$. (b) Bipartite mutual information scaling for three L-CUBE configurations: sub-volume ($\beta = \log(3)/\log(4) \approx 0.79$), logarithmic, and constant. The sub-volume configuration matches the scaling regime observed in natural language, while the logarithmic and constant configurations enable testing architectural capacity across different information growth rates.

puts a real-valued sequence $w_{1:L}$ of length $L = n^D$. The construction is highly efficient, requiring $O(L)$ operations for sequence generation and conditional probability computation when $m < n$, and $O(L \log L)$ in the volume-law case $m = n$, enabling efficient scaling to arbitrarily long contexts as discussed in Appx. C.IV.

**Multi-scale structure.** The construction proceeds through $D$ scales, where scale $d$ represents one level of compositional hierarchy. At each scale, the generator performs three operations: (i) $m$ "reused" channels carry information from the previous (coarser) scale while $n - m$ fresh Gaussian innovations inject new information; (ii) block correlator $\mathcal{M}^{(d)}$ combines these within local blocks; (iii) flattening produces a finer-resolution sequence. The optional local correlation operation (Alg. LOCALCORRELATE) smooths artificial discontinuities at block boundaries, analogous to transition words or bridging sentences in natural text that maintain cohesion across structural boundaries.

**Reuse mechanism and long-range dependencies.** The reuse parameter $m$ determines how many independent information pathways connect distant positions through the hierarchy. When $m$ channels are reused, each position at scale $d$ influences $m$ distinct positions at scale $d+1$, creating a branching structure that is the *sole* source of long-range dependencies and directly controls the scaling of $I^{\mathrm{BP}}$. Larger $m$ corresponds to richer compositional structure, analogous to how more aspects of a sentence's meaning carry over into the following text.

**Exact tractability.** The construction is linear-Gaussian by design. All conditional distributions are therefore Gaussian: $p(w_t \mid w_{1:t-1}) = \mathcal{N}(\mu_t, \sigma_t^2)$, where the parameters $(\mu_t, \sigma_t)$ can be computed exactly by propagating sufficient statistics through the same hierarchical structure shown in Alg. LCUBEGENERATOR. This enables ground-truth evalu-

---

**Algorithm 1** EXPANDONCE: Single scale expansion

**Require:** reused channels $X \in \mathbb{R}^{B \times A \times m}$, block size $n$, reuse $m$, scale map $\mathcal{M}^{(d)} \in \mathbb{R}^{n \times n}$
**Ensure:** expanded sequence $W \in \mathbb{R}^{B \times (An)}$
1: sample $E \sim \mathcal{N}(0,1)^{B \times A \times (n-m)}$
2: $V \leftarrow \mathrm{concat}(X, E)$      // $V \in \mathbb{R}^{B \times A \times n}$
3: $Z \leftarrow \mathcal{M}^{(d)}V$    // block correlate along last axis
4: **return** reshape($Z$, $B \times (An)$)    // flatten to create reuse structure

---

**Algorithm 2** LOCALCORRELATE (optional)

**Require:** sequence $W \in \mathbb{R}^{(B) \times (An)}$, block size $n$, current coarse length $A$, scale index $d$
**Ensure:** updated $W$
1: **for** $t = 1$ **to** $A - 1$ **do**
2:    compute $2 \times 2$ map $C_t^{(d)}$ from Gaussian second moments
3:    apply $C_t^{(d)}$ to boundary pair $(W_{tn}, W_{tn+1})$
4: **end for**
5: **return** $W$

---

ation of how well models capture the dependencies, which is further explained in Appx. C.III.

### 4.3. Information scaling and connection to natural language

The reuse parameter $m$ controls how many independent information channels can influence both sides of a long-range bipartition, thereby determining the scaling of $I^{\mathrm{BP}}$ as $L = n^D$ grows. A key observation is that sub-volume scaling requires $1 < m < n$: when multiple channels are reused, coarser hierarchical levels contribute proportionally more information across bipartitions than finer levels, with

---

**Algorithm 3** LCUBEGENERATOR

---

**Require:** block size $n$, reuse $m \leq n$, depth $D$, per-scale maps $\{\mathcal{M}^{(d)}\}_{d=1}^{D}$, optional local correlation flag
**Ensure:** $W \in \mathbb{R}^L$ with $L = n^D$
 1: $A \leftarrow 1$, $B \leftarrow m^{D-1}$
 2: sample $X \sim \mathcal{N}(0,1)^{B \times A \times m}$
 3: **for** $d = 1$ **to** $D$ **do**
 4:     $W \leftarrow$ EXPANDONCE$(X, n, m, \mathcal{M}^{(d)})$     // Alg. EXPANDONCE
 5:     **if** local correlation enabled **then**
 6:        $W \leftarrow$ LOCALCORRELATE$(W, n, A, d)$     // Alg. LOCALCORRELATE
 7:     **end if**
 8:     $A \leftarrow An$
 9:     **if** $d < D$ **then**
10:        reshape $X \leftarrow$ reshape$(W, \ (B/m) \times A \times m)$; $B \leftarrow B/m$
11:     **end if**
12: **end for**
13: **return** $W_{1,:}$ // after $D$ scales, $B = 1$ (single sequence)

---

the exponent $\beta = \log m / \log n$ quantifying this asymmetry.

Each additional scale creates $m$ new pathways for shared information to propagate across a split, leading to power-law scaling $L^{\log m / \log n}$ in the intermediate regime $1 < m < n$. The extremal cases recover simpler behaviors: $m = 1$ yields constant scaling, while $m = n$ produces volume-law scaling. When $m = 1$ with local correlation enabled (Alg. LOCALCORRELATE), weak cross-scale coupling produces logarithmic growth.

**Proposition 4.1** (Scaling regimes (informal)). *Let $L = n^D$, fix $0 < \alpha < 1$, and set $\ell = \lfloor \alpha L \rfloor$. With suitably chosen $\{\mathcal{M}^{(d)}\}_{d=1}^{D}$, the bipartite mutual information $I_{\ell;L}^{\mathrm{BP}}$ exhibits: (i) constant: $m = 1$, $I_{\ell;L}^{\mathrm{BP}} = O(1)$; (ii) logarithmic (with local correlation at $m = 1$): $I_{\ell;L}^{\mathrm{BP}} = O(\log L)$; (iii) sub-volume: $1 < m < n$, $I_{\ell;L}^{\mathrm{BP}} = \Theta(L^\beta)$ with $\beta = \frac{\log m}{\log n}$; (iv) volume-law: $m = n$, $I_{\ell;L}^{\mathrm{BP}} = \Theta(L)$.*

See Appx. C.V for detailed proofs. Beyond matching the bipartite mutual information scaling observed in natural language, L-CUBE also reproduces the power-law decay of two-point mutual information $I(W_i; W_j)$ with distance $|i - j|$ that has been documented in natural language corpora as shown in Appx. C.VII. Details of the generator parameterization, including the block correlation maps $\mathcal{M}^{(d)}$ (Appx. C.I) and the optional local correlation operation (Appx. C.II), are provided in Appx. C.

## 5. Benchmarking Results

We evaluate how different architectural families capture hierarchical long-range dependencies using L-CUBE. Our benchmark includes transformers with full attention (GPT-2, GPT-NeoX, Qwen3) (Radford et al., 2019; Black et al., 2022; Biderman et al., 2023; Yang et al., 2025), state space models (Mamba, Mamba2, RWKV-7) (Gu & Dao, 2024; Dao & Gu, 2024; Peng et al., 2025), log-linear attention (LLA) (Guo et al., 2025), and sliding-window attention (Mistral with varying window sizes) (Jiang et al., 2023).

**Experimental setup.** We train models on two sequence lengths: the full sequence of length $L$ (corresponding to $X \cup Y$ in the theory) and a shorter sequence of length $L/4$ (corresponding to $Y$ alone). This design exploits the self-similarity of the hierarchical generator and allows efficient reuse of training configurations across different evaluation lengths. Models are adapted to the continuous-valued generator without tokenization or quantization: token embeddings are replaced with continuous input adapters and the softmax head with Gaussian output heads that predict $(\hat{\mu}_t, \hat{\sigma}_t)$, preserving attention, recurrent updates, and positional encoding. All architectures are trained on an unbounded stream of fresh samples from the generator under the same optimization recipe and run to convergence, so the comparisons isolate architectural effects from token-budget and FLOP-budget confounds. See Appx. E for experiment details. We measure long-context utilization $U$, which quantifies how much of the available bipartite mutual information the model successfully extracts, and test whether L-CUBE data can augment pretraining when natural long-context data is limited. We focus on representative results in the main paper; Appx. F contains extensive additional experiments including explicit KL divergence measurements and detailed analysis of the relative residual $\rho$; Appx. G extends the evaluation to the logarithmic MI scaling regime; and Appx. H contains additional augmentation experiments.

### 5.1. Long-context utilization across architectures

Fig. 3 shows long-context utilization as a function of sequence length for different architectural families. The black dashed line indicates the ground-truth bipartite mutual information $I_{3L/4;L}^{\mathrm{BP}}$ from the L-CUBE generator, which represents the available information gain from context.

Full-attention transformers (panel a) track the ground truth across all tested sequence lengths up to $16{,}384$. GPT-2, GPT-NeoX, and Qwen3 all successfully extract the hierarchical long-range dependencies, consistent with the $L^2M$ theory's prediction that transformers have sufficient history state capacity for sub-volume scaling regimes.

SSMs (panel b) initially track the ground truth but then peak and subsequently decline as sequence length increases, exhibiting the failure mode predicted by $L^2M$ theory: fixed history state dimension cannot capture indefinitely growing bipartite mutual information. Within the SSM family, RWKV-7 (163M parameters) maintains utilization best, fol-

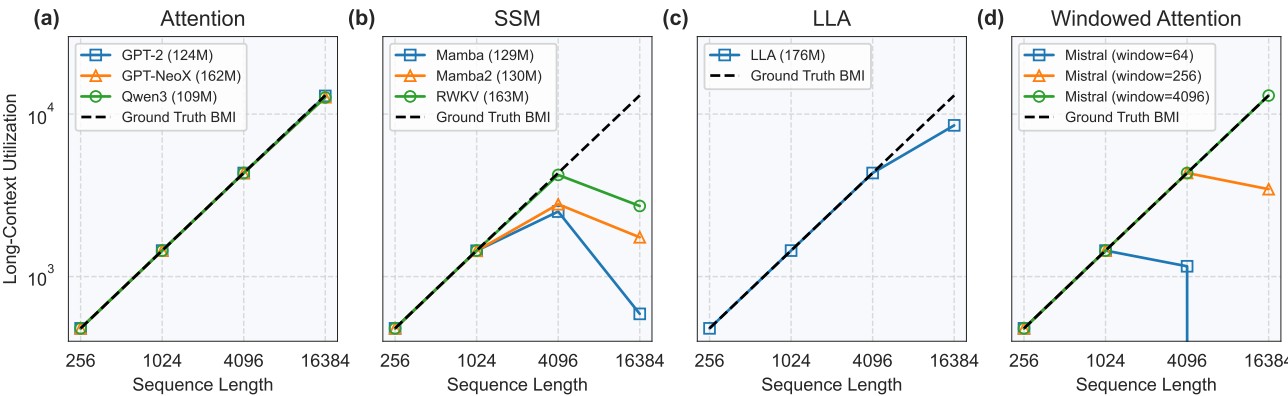

*Figure 3.* Long-context utilization $U$ versus sequence length for models with approximately 100-200M parameters. Panel (d) uses a 430M model but is included to compare window-size effects. The black dashed line indicates ground-truth bipartite mutual information $I^{\mathrm{BP}}_{3L/4;L}$ from the L-CUBE generator. (a) Full-attention transformers (GPT-2, GPT-NeoX, Qwen3) track the ground truth across all tested lengths. (b) State space models (Mamba, Mamba2, RWKV) degrade as sequence length increases, exhibiting the failure mode predicted by fixed history state capacity. (c) Log-linear attention (LLA) performs better than SSMs but fails to capture sub-volume scaling at longer lengths. (d) Sliding-window attention (Mistral) tracks ground truth within the context window but exhibits sharp degradation when context exceeds the context window size.

lowed by Mamba2 (130M) and Mamba (129M). The gap between model curves and the ground truth corresponds to the KL residual $\Delta_{\mathrm{KL}}$, which measures how the model fails to capture dependencies.

LLA (panel c) performs better than SSMs due to its logarithmic history state growth, but still fails to match the ground truth at longer lengths. This demonstrates that logarithmic scaling, while better than constant, is insufficient for capturing the sub-volume power-law growth in our benchmark. The widening gap confirms the $\mathrm{L^2M}$ prediction: the model's history state dimension must scale at least as fast as $I^{\mathrm{BP}}$ for effective long-context modeling. Appx. G evaluates the same architectures in the logarithmic MI scaling regime.

Mistral with sliding window attention (panel d) tracks the ground truth while context remains within the window, but exhibits sharp degradation when sequence length exceeds the context window size. The transition occurs when relevant context begins to fall outside the fixed window, demonstrating the hard limitation of finite attention span.

### 5.2. Scaling model capacity

Fig. 4 examines how model size affects long-context utilization for transformers and SSMs at smaller scales. While $\mathrm{L^2M}$ theory places no fundamental limit on transformers' ability to capture long-range dependencies, we observe that smaller transformers can fail in practice. Interestingly, GPT-2 (64M) maintains better long-context utilization than GPT-NeoX (70M) despite similar parameter counts. We attribute this to architectural differences in positional encoding: GPT-NeoX uses vanilla rotary embeddings (Su et al., 2024), without extrapolation methods such as YaRN. Despite RoPE's intended support for context-length extrapolation, it may

limit a transformer's ability to capture long-range dependencies when the hidden dimension is small. More broadly, implementation choices such as positional encoding may shrink an architecture's *effective* history state size below the asymptotic dimension counted by $\mathrm{L^2M}$ theory, a gap L-CUBE surfaces empirically.

For SSMs (panel b), increasing model size improves long-context utilization and delays the peak. The improvement is more pronounced for Mamba2 (785M) than Mamba (793M), with Mamba2 showing almost no decline within our tested range, likely due to Mamba2's more expressive state update mechanism.

Additional results of various model sizes up to 1.4B can be found in Appx. F.

### 5.3. Parameter and compute efficiency

Fig. 5 analyzes the relationship between model resources and long-context capability. We define *maximum utilized context length* as the context length (i.e., length of $\boldsymbol{X}$, which is $3L/4$) of the longest sequence length where the model achieves relative residual $\rho < 1\%$, meaning it captures at least 99% of available information. Our longest tested sequence length corresponds to context length 12,288; models reaching this value may be capable of utilizing even longer contexts.

Panel (a) shows that transformers achieve substantially longer utilized context per parameter compared to other architectures. For instance, a 124M GPT-2 model utilizes context length 12,288, while a 1.4B Mamba model peaks around context length 3,072. This dramatic difference reflects the fundamental advantage of transformers'

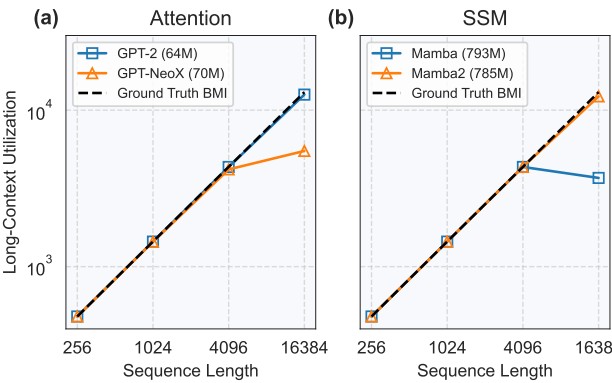

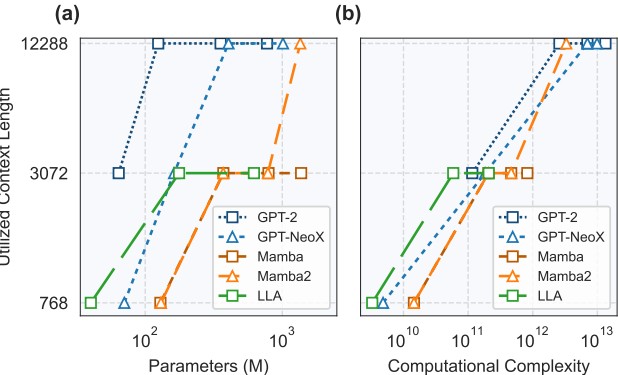

*Figure 4.* Long-context utilization versus sequence length for smaller models. (a) Transformers: GPT-2 (64M) maintains better utilization than GPT-NeoX (70M) despite similar parameter counts, likely due to differences in positional encoding (learned vs. rotary embeddings). (b) State space models: Mamba (793M) and Mamba2 (785M) show that increasing model size delays but cannot prevent the eventual plateau, confirming the fundamental architectural limitation.

*Figure 5.* Maximum utilized context length (defined as the context length $3L/4$ for the longest evaluated sequence length achieving relative residual $\rho < 1\%$) versus model resources. Models reaching 12,288 tokens may be capable of utilizing even longer contexts. (a) Transformers achieve substantially longer utilized context per parameter than other architectures. (b) When plotted against theoretical computational complexity, all architectures exhibit similar slopes despite different parameter efficiencies, suggesting a potential universal compute-scaling law for long-context capability.

unbounded context integration over SSMs' fixed-capacity recurrent states.

Panel (b) plots the same data against theoretical computational complexity (FLOPs per forward pass). Remarkably, despite their very different parameter efficiencies, all architectures exhibit similar slopes when viewed through the lens of computation. This suggests a potential universal scaling law: the rate at which long-context capability improves with computational budget may be architecture-independent, even though the efficiency with which architectures convert parameters into computation varies dramatically. We note that the constant offset between curves should be ignored due to differences in theoretical FLOP counts versus actual implementations.

We refer to Appx. F for complete benchmarking results.

### 5.4. Pretraining augmentation

Fig. 6 shows the effect of augmenting pretraining with L-CUBE sequences on length generalization. We train Pythia-410M from scratch on PG19 with primarily short sequences (256 tokens) and 1% long real data, with and without synthetic L-CUBE augmentation. Augmented models achieve lower loss when evaluated at 4096 tokens, where the evaluation length requires substantial extrapolation beyond the typical training length. Appx. H varies the fraction of long real data and the evaluation length.

These results demonstrate that L-CUBE's controllable hierarchical structure enables precise diagnosis of architectural limitations. The benchmark confirms $L^2M$ theory predictions, reveals practical failure modes even in theoretically sufficient architectures, and uncovers unexpected regulari-

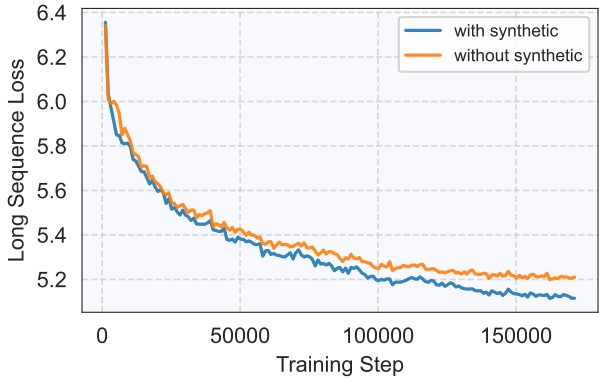

*Figure 6.* Long-sequence loss for Pythia-410M trained on PG19 with 1% long real data, evaluated at 4096 tokens, with and without L-CUBE augmentation.

ties in how different designs trade off parameters, computation, and long-context capability.

## 6. Discussion

Our central contribution is making explicit a distinction that has been implicit in sequence modeling: *what a model knows* versus *what long-range dependencies it can represent*. This separation has immediate practical value: when a model fails on long sequences, L-CUBE determines whether the failure is capacity-limited or knowledge-limited, making architectural decisions empirically testable without full-scale training on real data (Appx. I provides practical guidelines for using L-CUBE in architecture development).

L-CUBE acts as a capacity filter rather than a performance predictor. Real-world long-context performance also de-

pends on world knowledge, reasoning, and other capabilities, and knowledge itself can live at long range (Modarressi et al., 2025); an architecture that passes L-CUBE may still fail downstream for these other reasons, while an architecture that fails L-CUBE will struggle on any task requiring long-range structure.

Among synthetic constructions, Markov chains lack long-range dependencies by definition, and hierarchical constructions (Lin & Tegmark, 2017) match the two-point correlations of natural language but not its bipartite mutual information scaling, the quantity that governs long-range modeling capacity (Chen et al., 2025). L-CUBE provides exact tractability together with controllable bipartite scaling matched to natural language.

Our experiments validate $L^2M$ capacity predictions empirically: capacity-sufficient architectures like full-attention transformers maintain high utilization across lengths, while capacity-limited architectures like fixed-state SSMs show declining utilization as predicted. Interestingly, when viewed through computational budget, all architectures exhibit similar scaling rates in utilized context length despite dramatic differences in parameter efficiency [Fig. 5 (b)], suggesting that long-context capability may be governed by a compute-scaling regularity that transcends architectural details.

The knowledge-capacity separation applies beyond language to any domain with long-range dependencies, e.g. DNA sequences, music, program code, time series data. L-CUBE's generator can be configured to match domain-specific $I^{BP}$ scaling, enabling architectural diagnosis before expensive domain training. Limitations are discussed in Appx. J.

## 7. Conclusion

We have introduced L-CUBE, a benchmark that isolates architectural capacity for long-range dependencies from semantic knowledge through synthetic sequences with controllable hierarchical structure and exact ground-truth conditionals. The utilization metric quantifies how well architectures exploit available long-range signal as context grows, providing a principled capacity test independent of domain knowledge. Experiments across transformers, state space models, and efficient attention variants confirm that sufficiently large full-attention transformers track ground-truth mutual information across all tested lengths, while fixed-capacity architectures peak and decline as $L^2M$ theory predicts.

The results enable practitioners to test whether a particular architecture will maintain long-context capability at target sequence lengths before expensive training, and provide designers with direct feedback on how design choices

affect long-range capacity. Preliminary results suggest augmenting pretraining with L-CUBE data improves long-context performance when natural long-context data is limited. Future work includes extending the framework to non-stationary dependencies, developing hybrid architecture diagnostics, and investigating optimal curricula combining synthetic and real data.

## Acknowledgements

The authors acknowledge support from the National Science Foundation under Cooperative Agreement PHY-2019786 (The NSF AI Institute for Artificial Intelligence and Fundamental Interactions) and the MIT Generative AI Impact Consortium. Z.C. acknowledges support from the MathWorks Fellowship and partial support from the Henry W. Kendall (1955) Fellowship Fund. Z.C. and O.M. thank Amazon Web Services account team, including Brian McCarthy and Jared Novotny, for technical support. The research was sponsored by the United States Air Force Research Laboratory and the Department of the Air Force Artificial Intelligence Accelerator and was accomplished under Cooperative Agreement Number FA8750-19-2-1000. The computations in this paper were partly run on the FASRC cluster supported by the FAS Division of Science Research Computing Group at Harvard University. This research used the DeltaAI advanced computing and data resource, which is supported by the National Science Foundation (award OAC 2320345) and the State of Illinois, through allocation CIS240904 from the Advanced Cyberinfrastructure Coordination Ecosystem: Services & Support (ACCESS) program, supported by National Science Foundation grants #2138259, #2138286, #2138307, #2137603, and #2138296, and through the National Artificial Intelligence Research Resource (NAIRR) Pilot NAIRR250043.

## Impact Statement

This paper presents work whose goal is to advance the field of Machine Learning through improved benchmarking methodology for long-context language models. L-CUBE enables more principled architecture development by isolating capacity requirements from knowledge acquisition. We do not anticipate specific negative societal consequences beyond those inherent to advances in language model capabilities.

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

# A. Additional Related Work

**Long-context benchmarks (application-centric).**    Prominent evaluation suites including SCROLLS (Shaham et al., 2022), LongBench (Bai et al., 2024), and $\infty$Bench (Zhang et al., 2024), along with complementary efforts such as L-Eval (An et al., 2024), LooGLE (Li et al., 2024), LV-Eval (Yuan et al., 2025), Loong (extended multi-doc QA) (Wang et al., 2024), and XL$^2$Bench (Ni et al., 2024), evaluate long-context LLMs on realistic downstream tasks: long-document QA, multi-document QA, summarization, dialogue history, and code completion. These benchmarks measure end-to-end performance on practical tasks, though they necessarily entangle architectural capacity with semantic knowledge, memorization, and domain coverage. L-CUBE isolates long-range dependency utilization by removing semantic and vocabulary confounders, enabling evaluation against ground-truth conditionals.

**Benchmarking extremely long contexts and million-token regimes.**    As models claim context windows exceeding 100K-1M tokens, benchmarks have extended evaluation lengths through both realistic and synthetic constructions. $\infty$Bench (Zhang et al., 2024) explicitly targets contexts beyond 100K tokens, LongBench v2 extends to much larger raw contexts while emphasizing deeper reasoning under realistic multitasks (Bai et al., 2025), and NeedleBench studies retrieval-and-reasoning behavior at million-token scales (Li et al., 2025). These efforts highlight that supporting ultra-long inputs does not guarantee robust utilization across positions or with distributed evidence. L-CUBE directly controls the long-range information budget via $I^{\mathrm{BP}}$ scaling and measures how much predictive information is extracted.

**Retrieval and position-sensitivity probes.**    Needle-in-a-haystack tests (Kamradt, 2023) remain common probes, and RULER systematizes retrieval-style and beyond-retrieval synthetic tasks with flexible length and difficulty (Hsieh et al., 2024). More broadly, *Lost in the Middle* shows strong position sensitivity: models often underuse evidence placed in context middles even when claiming long-context capability (Liu et al., 2024). These probes evaluate access to specific content at known positions. Retrieval of localized information can be separable from exploiting distributed long-range statistical dependencies; our benchmark targets the latter by construction and scores models by utilization of predictive information rather than discrete retrieval success.

**Evaluation frameworks and meta-benchmarks.**    Recent work emphasizes that long-context evaluation is noisy and benchmark-dependent. HELMET proposes comprehensive application-centric evaluation and analyzes why single probes like needle-in-a-haystack can be misleading (Yen et al., 2025). LOOM-Scope argues for standardized settings across diverse benchmarks and efficient evaluation workflows (Tang et al., 2025). These frameworks identify a fundamental challenge in long-context evaluation: separating architectural capacity from semantic knowledge and dataset coverage. Controlled tests with known ground-truth can reduce ambiguity in model comparison and architectural development.

**Synthetic and controlled benchmarks.**    Controlled suites such as Long Range Arena (Tay et al., 2021) and algorithmic tasks like ListOps (Nangia & Bowman, 2018) provide diagnostics under simplified distributions. However, these benchmarks often do not reproduce the information-theoretic structure characteristic of natural language, specifically, the sub-volume growth of predictive information with sequence length. L-CUBE generates sequences with controllable $I^{\mathrm{BP}}$ scaling that matches natural text ($I^{\mathrm{BP}} \sim L^{0.8}$) while maintaining exact tractability, enabling direct theory-to-experiment connections.

**Associative recall and literal-matching probes.**    Synthetic, knowledge-free diagnostics based on associative recall (Arora et al., 2024; Poli et al., 2023) and lexically decoupled needle-in-a-haystack (Modarressi et al., 2025) share L-CUBE's goal of isolating architectural capacity from semantic knowledge, and report architectural rankings consistent with our transformer versus state space model results. They score discrete retrieval or association ability rather than the continuous, distributed long-range dependencies of natural language; L-CUBE targets the latter through exact ground-truth conditionals, controllable $I^{\mathrm{BP}}$ scaling matched to natural text, and a direct connection to L$^2$M capacity theory.

**Architectures for efficient long-context modeling.**    Various architectural approaches target the quadratic complexity of standard attention mechanisms. Innovations include sparse attention patterns (Child et al., 2019; Ding et al., 2023; Beltagy et al., 2020; Zaheer et al., 2020), recurrent mechanisms (Dai et al., 2019; Rae et al., 2020; Sukhbaatar et al., 2019), and alternative formulations such as linear attention (Katharopoulos et al., 2020), state space models (Gu et al., 2022; Gu & Dao, 2024; Dao & Gu, 2024), and hybrid architectures (He et al., 2025; Peng et al., 2023; Sun et al., 2025; Beck et al., 2024; De et al., 2024). Memory-efficient implementations like Flash Attention (Dao et al., 2022; Dao, 2024; Shah et al., 2024), Lightning Attention (Qin et al., 2024), and Paged Attention (Kwon et al., 2023) improve computational efficiency

while maintaining the underlying complexity scaling. These architectural innovations address computational and memory constraints, while L-CUBE provides a capacity-focused diagnostic to evaluate whether proposed architectures can actually extract the required amount of long-range predictive information.

**Training recipes and context extension methods.**   Beyond evaluation, extensive work studies extending context windows via continued pretraining, data curricula, and positional extrapolation. Research explores effective long-context scaling through continual pretraining in foundation-model settings (Xiong et al., 2024) and data engineering recipes for scaling to 128K contexts (Fu et al., 2024). Studies like ProLong investigate how to train long-context models effectively, including evaluation protocols and supervised fine-tuning considerations (Gao et al., 2025). L-CUBE serves as a diagnostic benchmark for these architectural and training choices, enabling controlled capacity testing before expensive deployment-oriented training.

**Effective context length and utilization gaps.**   Even with long training lengths and large claimed windows, models may effectively use only a fraction of available context. Work analyzing why effective context length falls short attributes limitations to position distribution skew and proposes inference-time interventions (An et al., 2025). This line of analysis aligns with the emphasis on measuring utilization rather than nominal context length. L-CUBE provides a capacity-focused setting to quantify utilization as long-range information demands increase through the KL residual metric.

**Information theory and physics-inspired approaches.**   Information-theoretic principles and physics-inspired approaches have guided machine learning development across multiple dimensions (Tishby & Zaslavsky, 2015; Goldfeld & Polyanskiy, 2020; Chen & Luo, 2024), leading to novel architectures (Luo & Clark, 2019; Luo et al., 2023; Wang et al., 2021; Chen et al., 2022; Lami et al., 2022; Wu et al., 2023; Chen et al., 2023; Dugan et al., 2024), training methods (Stokes et al., 2020; Chen et al., 2024b;a), and broad applications (Carleo et al., 2019; Lee et al., 2020; Luo et al., 2022; Moro et al., 2025; Chen et al., 2024b; Choi et al., 2024). The $L^2M$ theory (Chen et al., 2025) specifically formalizes long-context capacity constraints via history-state bounds linked to bipartite mutual information. L-CUBE continues this direction by operationalizing theoretical predictions: constructing a benchmark where $I^{\mathrm{BP}}$ is exactly controllable and ground-truth conditionals are available, enabling direct measurement of conditional mismatch and establishing concrete connections between information-theoretic capacity requirements and empirical architectural performance.

# B. Continuous Variables and Differential Entropy

Our benchmark uses continuous Gaussian sequences. While differential entropy is not invariant under coordinate transformations and can be negative, our metrics are constructed from KL divergences and density ratios, which are invariant.

**Proposition B.1.** *KL divergence is invariant under invertible transformations.*

*Proof.* Under transformation $Y' = f(Y)$ with Jacobian $J$, densities transform as $p_{Y'}(y') = p_Y(f^{-1}(y'))|J^{-1}|$ and $q_{Y'}(y') = q_Y(f^{-1}(y'))|J^{-1}|$. The KL divergence becomes

$$D_{\mathrm{KL}}(p_{Y'}\|q_{Y'}) = \int p_{Y'}(y') \log \frac{p_{Y'}(y')}{q_{Y'}(y')} dy' \tag{B.1}$$

$$= \int p_{Y'}(y') \log \frac{p_Y(f^{-1}(y'))|J^{-1}|}{q_Y(f^{-1}(y'))|J^{-1}|} dy' \tag{B.2}$$

$$= \int p_{Y'}(y') \log \frac{p_Y(f^{-1}(y'))}{q_Y(f^{-1}(y'))} dy'. \tag{B.3}$$

The Jacobian terms cancel in the ratio. By change of variables with $y = f^{-1}(y')$ and $dy = |J^{-1}|dy'$, this equals $D_{\mathrm{KL}}(p_Y\|q_Y)$. $\square$

Our primary metrics are defined entirely as sums and differences of KL divergences. The KL residual is

$$\Delta_{\mathrm{KL}}(X;Y) = D_{\mathrm{KL}}(p_{Y|X}\|q_{xy}) - D_{\mathrm{KL}}(p_Y\|q_y). \tag{B.4}$$

The available gain is the mutual information under $p$, which can be written as a KL divergence:

$$G(X;Y) = I_p(X;Y) = D_{\mathrm{KL}}(p_{XY}\|p_X \otimes p_Y) = \mathbb{E}_{p_X}[D_{\mathrm{KL}}(p_{Y|X}\|p_Y)], \tag{B.5}$$

where $p_X \otimes p_Y$ denotes the product distribution.

The utilization decomposes as

$$U(X;Y) = G(X;Y) - \Delta_{\mathrm{KL}}(X;Y) = \mathbb{E}_{p_X}[D_{\mathrm{KL}}(p_{Y|X}\|p_Y)] - D_{\mathrm{KL}}(p_{Y|X}\|q_{xy}) + D_{\mathrm{KL}}(p_Y\|q_y). \qquad \text{(B.6)}$$

Since all three metrics are expressed purely as sums and differences of KL divergences, and KL divergence is invariant under coordinate transformations (previous proposition), it follows immediately that $\Delta_{\mathrm{KL}}(X;Y)$, $G(X;Y)$, and $U(X;Y)$ are all coordinate-invariant. Despite using continuous variables with coordinate-dependent differential entropies, our metrics involve only well-defined, invariant quantities.

# C. Generator Implementation

This section provides complete implementation details for the L-CUBE hierarchical generator introduced in Sec. 4. We explain the block correlation parameterization, conditional probability computation, and computational complexity analysis.

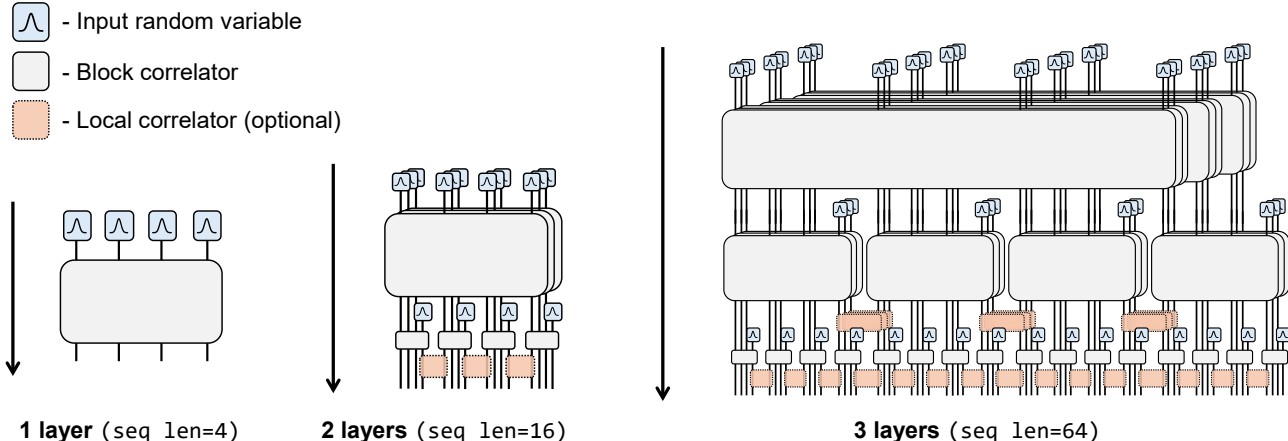

*Figure C.1.* Hierarchical structure of the L-CUBE generator. Information flows from coarser scales (top) to finer scales (bottom) through the reuse mechanism. At each scale $d$, $m$ channels are reused from the previous scale while $n - m$ fresh Gaussian innovations are injected, then mixed by block correlator $\mathcal{M}^{(d)}$ and flattened to create branching structure.

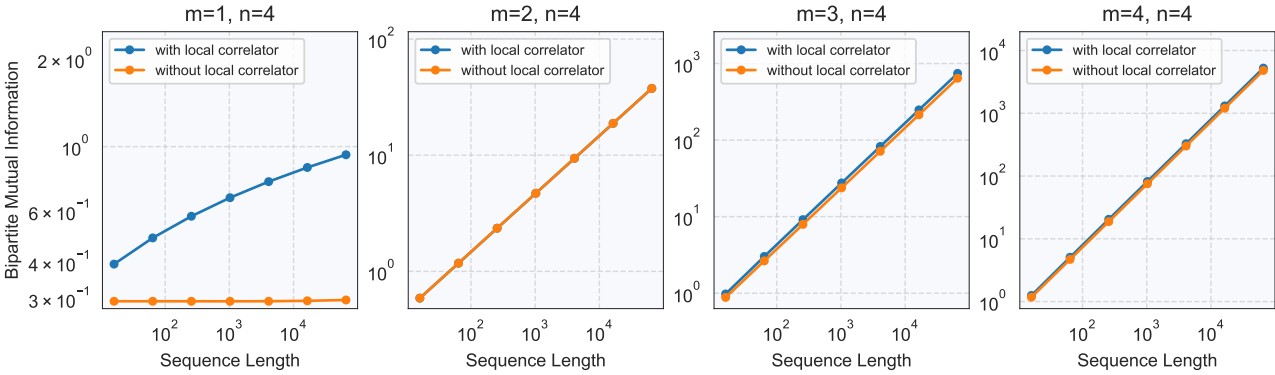

*Figure C.2.* Bipartite mutual information for $n = 4$ with varying $m$. Without local correlation: $m = 1$ gives constant scaling, $m = 2$ gives $\beta = 0.5$, $m = 3$ gives $\beta \approx 0.79$, $m = 4$ gives volume-law. With local correlation, $m = 1$ exhibits logarithmic growth.

## C.I. Block Correlation Parameterization

The block correlators $\mathcal{M}^{(d)} \in \mathbb{R}^{n \times n}$ control how information mixes within blocks at each scale. These linear transformations determine the covariance structure within blocks and, combined with the reuse mechanism, control the overall scaling of bipartite mutual information.

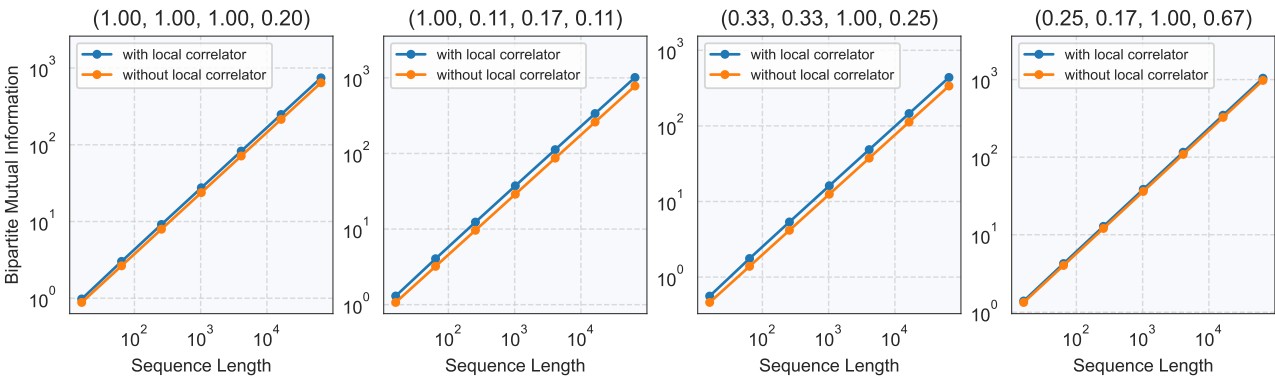

*Figure C.3.* Effect of variance parameters on $I^{\mathrm{BP}}$ for $m = 3$, $n = 4$. Different unnormalized diagonal eigenvalue configurations produce similar sub-volume scaling, demonstrating robustness of the construction to parameter choices.

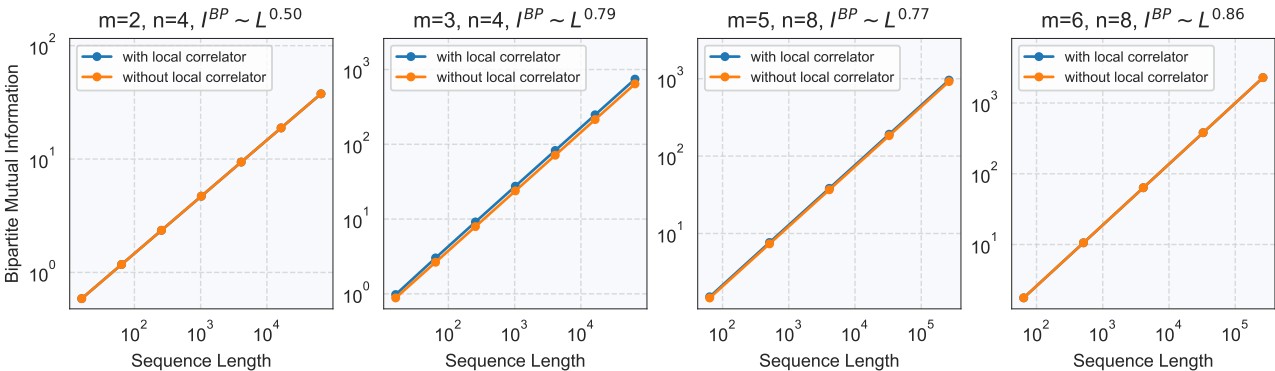

*Figure C.4.* Scaling for different $(m, n)$ combinations. Each configuration exhibits sub-volume law with exponent $\beta = \log m / \log n$, confirming the theoretical characterization in Sec. C.V.

The correlators can be specified by any desired output covariance matrix $\Sigma^{(d)}$, from which $\mathcal{M}^{(d)} = (\Sigma^{(d)})^{1/2}$ is obtained via matrix square root or Cholesky decomposition. We use a Hadamard-basis parameterization for interpretability and numerical stability. For block size $n = 2^k$, the $2^k \times 2^k$ Hadamard matrix $H_k$ is defined recursively:

$$H_1 = \begin{bmatrix} 1 & 1 \\ 1 & -1 \end{bmatrix}, \quad H_k = \begin{bmatrix} H_{k-1} & H_{k-1} \\ H_{k-1} & -H_{k-1} \end{bmatrix}. \tag{C.1}$$

The normalized matrix $Q = H_k / \sqrt{2^k}$ is orthogonal, satisfying $Q^\top Q = I$. We parameterize

$$\mathcal{M}^{(d)} = Q \cdot \mathrm{diag}(\sqrt{\lambda_1^{(d)}}, \ldots, \sqrt{\lambda_n^{(d)}}), \tag{C.2}$$

where variance parameters satisfy $\lambda_i^{(d)} \geq 0$ and $\sum_{i=1}^n \lambda_i^{(d)} = n$.

This parameterization has two advantages. First, the diagonal variance parameters directly control variance allocation across orthogonal Hadamard modes, providing interpretable control over correlation structure. Second, the normalization constraint $\sum_i \lambda_i^{(d)} = n$ transparently ensures bounded variance across scales: if inputs have identity covariance, outputs have covariance

$$\Sigma_{\mathrm{out}} = \mathcal{M}^{(d)}(\mathcal{M}^{(d)})^\top = Q \cdot \mathrm{diag}(\lambda_1^{(d)}, \ldots, \lambda_n^{(d)}) \cdot Q^\top, \tag{C.3}$$

with trace $\mathrm{tr}(\Sigma_{\mathrm{out}}) = \sum_i \lambda_i^{(d)} = n$, maintaining $O(1)$ average variance per coordinate.

Our default configuration targets sub-volume scaling with exponent $\beta \approx 0.79$ to match natural language. We use block size $n = 4$, reuse parameter $m = 3$ (giving $\beta = \log(3)/\log(4) \approx 0.793$), and depth $D \in \{3, 4, 5, 6, 7\}$ for sequence lengths $L \in \{64, 256, 1024, 4096, 16384\}$. The unnormalized variance parameters $[1, 1, 1, 0.2]$ are normalized to sum $n = 4$,

providing moderate variance concentration on the first few Hadamard modes while maintaining good mixing and numerical stability. Unless otherwise stated, the default sub-volume experiments use local correlation enabled. Fig. C.3 demonstrates that varying these variance parameters produces similar sub-volume scaling, confirming robustness of the construction to this choice. Different $(m, n)$ combinations yield different power-law exponents as predicted by the theoretical characterization (Fig. C.4).

Figures C.2-C.4 demonstrate empirically that the reuse parameter $m$ controls the scaling exponent as predicted by Proposition 4.1.

### C.II. Local Correlation

The local correlation operation addresses artificial discontinuities at block boundaries created by flattening the hierarchical structure. Without this operation, positions within blocks have correlated structure from the block correlator $\mathcal{M}^{(d)}$, while positions across block boundaries only inherit correlations from the higher scale, creating sharp discontinuities in the correlation function.

At scale $d$, after flattening to length $A_d \cdot n$ where $A_d = n^{d-1}$, there are $A_d - 1$ boundaries between adjacent blocks. For each boundary, we apply a linear transformation to the pair of positions straddling the boundary that adjusts their joint covariance to match the covariance structure of within-block neighbors. The transformation preserves marginal variances while introducing correlation between the boundary pair, ensuring smooth decay of the two-point correlation function $\mathrm{Cov}(W_i, W_j)$ as a function of distance $|i - j|$.

The strength of introduced correlation is tuned so that boundary pairs have similar covariance eigenvalues to adjacent within-block pairs. This ensures that the hierarchical structure does not create artificial correlation jumps at block boundaries when the sequence is arranged in one-dimensional order. The result is a smooth power-law decay of two-point mutual information with distance, matching the behavior observed in natural language.

For $m = 1$, local correlation transforms constant scaling into logarithmic scaling by accumulating $\Theta(1)$ nats of cross-boundary information at each of $D = \log_n L$ scales. For $m > 1$, local correlation does not change the asymptotic $\Theta(L^\beta)$ scaling but improves smoothness of correlations across boundaries, as demonstrated in Fig. C.2.

### C.III. Conditional Probability Computation

The generator provides exact ground-truth conditionals $p(w_t \mid w_{1:t-1}) = \mathcal{N}(\mu_t, \sigma_t^2)$ by exploiting the linear-Gaussian structure of the hierarchical construction. Since all operations are linear transformations of Gaussian variables, all conditionals are Gaussian with parameters computable in closed form.

The hierarchical factorization enables efficient computation. Position $t$ at the finest scale $D$ has a structured dependence on earlier positions through the branching tree: it depends directly on $m$ positions at scale $D - 1$, which in turn depend on $m$ positions each at scale $D - 2$, and so forth. Rather than conditioning on all $t - 1$ previous positions, we can compute $p(w_t \mid w_{1:t-1})$ by propagating Gaussian sufficient statistics (means and covariances) through this hierarchical dependence structure.

At each scale $d$, the block correlator $\mathcal{M}^{(d)}$ transforms Gaussian variables with known parameters. Given the conditional parameters at scale $d$, the conditional parameters at scale $d + 1$ follow from the standard formulas for linear transformations of Gaussians. As positions are generated, we update conditional distributions using Gaussian conditioning:

$$p(w_{t+1} \mid w_{1:t}) = \mathcal{N}\left(\mu_{t+1} + \Sigma_{t+1,t}\Sigma_{t,t}^{-1}(w_t - \mu_t),\ \Sigma_{t+1,t+1} - \Sigma_{t+1,t}\Sigma_{t,t}^{-1}\Sigma_{t,t+1}\right), \tag{C.4}$$

where $\mu$ and $\Sigma$ denote the current conditional mean and covariance after conditioning on $w_{1:t-1}$, computed from the hierarchical structure.

This approach requires $O(1)$ operations per position when amortized over the sequence. For $m < n$, generating $w_{1:L}$ while computing all conditionals $p(w_t \mid w_{1:t-1})$ costs $O(L)$ time and $O(L)$ space; the volume-law case $m = n$ is $O(L \log L)$ as shown below. The implementation also computes reverse-order conditionals $p(w_t \mid w_{t+1:L})$ to enable efficient evaluation of arbitrary conditional probabilities.

## C.IV. Computational Complexity

**Theorem C.1** (Linear generation and evaluation). *The generator produces a sequence $w_{1:L}$ and computes all conditionals $p(w_t \mid w_{1:t-1})$ in $O(L)$ time and $O(L)$ space for $L = n^D$ when $m < n$, and $O(L \log L)$ time when $m = n$.*

*Proof.* The generator processes the sequence through $D$ scales. At scale $d$, the algorithm processes $A_d = n^{d-1}$ blocks, each of size $n$, applying the block correlator $\mathcal{M}^{(d)} \in \mathbb{R}^{n \times n}$ to each block. The cost at scale $d$ is

$$C_d = O(A_d n^2) = O(n^{d-1} \cdot n^2) = O(n^{d+1}). \tag{C.5}$$

However, the hierarchical construction generates multiple sequences in parallel. At scale $d$, we simultaneously generate $B_d = m^{D-d}$ independent sequences that will later branch further. Each requires the same $O(n^{d+1})$ operations, giving total cost at scale $d$:

$$C_d = O(B_d n^{d+1}) = O(m^{D-d} n^{d+1}). \tag{C.6}$$

Summing over all scales:

$$C_{\text{total}} = \sum_{d=1}^{D} O(m^{D-d} n^{d+1}) = O\left(n^2 \sum_{d=1}^{D} m^{D-d} n^{d-1}\right) \tag{C.7}$$

$$= O\left(n^2 m^{D-1} \sum_{d=1}^{D} m^{D-d+1-D} n^{d-1}\right) = O\left(n^2 m^{D-1} \sum_{d=1}^{D} m^{1-d} n^{d-1}\right) \tag{C.8}$$

$$= O\left(n^2 m^{D-1} \sum_{d=1}^{D} \left(\frac{n}{m}\right)^{d-1}\right). \tag{C.9}$$

For sub-volume scaling ($m < n$), this becomes:

$$\sum_{d=1}^{D} \left(\frac{n}{m}\right)^{d-1} = \frac{(n/m)^D - 1}{n/m - 1} = O\left(\left(\frac{n}{m}\right)^D\right). \tag{C.10}$$

Therefore:

$$C_{\text{total}} = O\left(n^2 m^{D-1} \cdot \left(\frac{n}{m}\right)^D\right) = O\left(n^2 m^{D-1} \cdot \frac{n^D}{m^D}\right) = O\left(\frac{n^{D+2}}{m}\right) = O(L), \tag{C.11}$$

where we use $L = n^D$ and absorb constants into $O(\cdot)$.

For volume-law scaling ($m = n$), the geometric series becomes an arithmetic series:

$$\sum_{d=1}^{D} 1 = D = \log_n L, \tag{C.12}$$

giving $C_{\text{total}} = O(n^2 n^{D-1} \cdot D) = O(n^{D+1} \log_n L) = O(L \log L)$.

Conditional probability computation is performed simultaneously during generation by maintaining Gaussian sufficient statistics (means and covariances). At scale $d$, there are $O(m^{D-d} n^d)$ positions being processed. Each position requires $O(1)$ operations for Gaussian conditioning updates, contributing $O(m^{D-d} n^d)$ to the cost at scale $d$. Summing over scales:

$$\sum_{d=1}^{D} O(m^{D-d} n^d) = O\left(m^D \sum_{d=1}^{D} \left(\frac{n}{m}\right)^d\right) = O\left(m^D \left(\frac{n}{m}\right)^D\right) = O(n^D) = O(L), \tag{C.13}$$

for $m < n$, which is dominated by the generation cost. The local correlation operation adds $O(n^{d-1})$ operations at scale $d$ for $A_d - 1$ boundaries, which is also dominated by the block correlator cost.

Space complexity is $O(L)$ as we store the final sequence and conditional parameters for each position. $\qquad\square$

## C.V. Scaling Regime Characterization

We characterize the scaling behavior of $I^{\mathrm{BP}}_{\ell;L}$ for the hierarchical generator with block size $n$, reuse $m$, depth $D$, and length $L = n^D$. Throughout we assume the block correlators $\mathcal{M}^{(d)}$ are chosen to maintain bounded variances and provide sufficient mixing within blocks. We explain the scaling regimes below.

**Proposition C.2** (Constant scaling). *When $m = 1$ without local correlation, for any $\ell \in \{1, \ldots, L - 1\}$, $I^{\mathrm{BP}}_{\ell;L} = I(W_{1:\ell}; W_{\ell+1:L}) = O(1)$.*

Consider a cut at position $\ell$, partitioning the sequence into left half $W_{1:\ell}$ and right half $W_{\ell+1:L}$. This cut corresponds to some location in the hierarchical construction. At some scale $d^*$, this cut passes between positions that are processed by a block correlator $\mathcal{M}^{(d^*)}$.

Above scale $d^*$ (at coarser scales $d < d^*$), the information that will eventually form the left half and right half originates from independent channels, since with $m = 1$ each position at a coarser scale influences exactly one position at the finer scale through reuse, and the $n - 1$ fresh innovations are independent. Therefore the left and right halves share no information from scales above $d^*$.

At scale $d^*$, at most one block correlator $\mathcal{M}^{(d^*)}$ can introduce correlation between the left and right halves (since with $m = 1$ each block processes independently and a cut intersects at most one block). This single $n \times n$ matrix operating on $n$-dimensional Gaussians with bounded variance introduces at most $O(1)$ nats of mutual information between the inputs that will later separate into left and right.

Below scale $d^*$ (at finer scales $d > d^*$), the operations are independent on the left versus the right: positions in the left half are only transformed using information from the left, and similarly for the right (Alg. EXPANDONCE applies $\mathcal{M}^{(d)}$ independently to each block). Since mutual information is invariant under independent transformations applied separately to each variable, the mutual information $I(W_{1:\ell}; W_{\ell+1:L})$ equals the mutual information introduced at scale $d^*$, which is $O(1)$.

Therefore $I^{\mathrm{BP}}_{\ell;L} = O(1)$ for any cut location $\ell$.

**Proposition C.3** (Sub-volume scaling). *When $1 < m < n$, for any fixed ratio $0 < \alpha < 1$ and $\ell = \lfloor \alpha L \rfloor$, $I^{\mathrm{BP}}_{\ell;L} = \Theta(L^\beta)$ where $\beta = \log m / \log n$.*

The key difference from the constant case is that with $m > 1$ reused channels, information at a coarser scale branches to multiple positions at the finer scale. This creates multiple parallel pathways connecting the left and right halves.

Consider a cut at ratio $\alpha$, i.e., $\ell = \lfloor \alpha L \rfloor$. At scale $d$, this cut intersects the sequence at some position within the $n^{d-1}$ coarse blocks at that scale. At this scale, multiple block correlators can contribute to the mutual information between left and right.

To count the number of contributing correlators, observe that at scale 1 there are $m$ initial channels. At each subsequent scale, each channel branches into $m$ positions (through the reuse mechanism in Alg. EXPANDONCE). Therefore at scale $d$, there are $m^{d-1}$ independent "lineages" of information flowing through the hierarchy. At scale $D$, a cut at ratio $\alpha$ will intersect roughly $\alpha \cdot m^D$ channels on the left and $(1 - \alpha) \cdot m^D$ channels on the right. However, what matters for mutual information is how many channels from scale 1 influence both sides.

More precisely, at the coarsest scale where the cut location becomes meaningful (which depends on $\alpha$ and the branching structure), there are $O(m^{D-1})$ block correlators operating in parallel on channels that will later separate into left and right. Each such correlator is an independent $n \times n$ matrix operating on independent Gaussian inputs (by construction of the generator, fresh innovations at each scale are independent across different branches). Each correlator contributes $\Theta(1)$ nats of mutual information.

Since these block correlators operate on independent inputs and produce independent outputs (across different branches), the mutual information contributions are additive:

$$I^{\mathrm{BP}}_{\ell;L} = \Theta(m^{D-1}). \tag{C.14}$$

Below this critical scale, transformations are again independent on left versus right, preserving mutual information. Using $L = n^D$, we have $D = \log_n L = \log L / \log n$, so:

$$m^{D-1} = m^{(\log L / \log n) - 1} = \frac{m^{\log L / \log n}}{m} = \frac{e^{(\log m / \log n) \log L}}{m} = \frac{L^{\log m / \log n}}{m} = \Theta(L^\beta), \tag{C.15}$$

where $\beta = \log m / \log n \in (0, 1)$ and the constant factor $1/m$ is absorbed into $\Theta(\cdot)$.

**Proposition C.4** (Volume-law scaling). *When $m = n$, for any fixed ratio $0 < \alpha < 1$ and $\ell = \lfloor \alpha L \rfloor$, $I_{\ell;L}^{\mathrm{BP}} = \Theta(L)$.*

The analysis is identical to the sub-volume case, except now $m = n$. At scale $D$, there are $n^{D-1}$ parallel block correlators contributing mutual information. Following the same calculation:

$$I_{\ell;L}^{\mathrm{BP}} = \Theta(n^{D-1}) = \Theta(n^{(\log L / \log n) - 1}) = \Theta\left(\frac{L}{n}\right) = \Theta(L). \tag{C.16}$$

This is volume-law scaling: mutual information grows linearly with sequence length.

**Local Correlation.** The local correlation operation (Alg. LOCALCORRELATE) applies transformations to pairs of adjacent positions across block boundaries at each scale. For $m = 1$, this introduces an additional logarithmic component to the mutual information. At each scale $d$, there are $O(n^{d-1})$ such pairs, but only $O(1)$ pairs contribute to mutual information between left and right at a given cut. Across $D = \log_n L$ scales, these contributions accumulate additively, yielding $I_{\ell;L}^{\mathrm{BP}} = \Theta(\log L)$ in addition to the constant baseline.

For $m > 1$, local correlation does not change the asymptotic scaling behavior, which remains dominated by the $\Theta(L^\beta)$ or $\Theta(L)$ contributions from the reuse mechanism. However, local correlation introduces meaningful short-range correlations between adjacent positions at each scale, improving smoothness of the correlation decay across the 1-dimensional sequence ordering.

### C.VI. Bipartite MI Properties

The bipartite mutual information $I_{\ell;L}^{\mathrm{BP}} = I(W_{1:\ell}; W_{\ell+1:L})$ depends on the cut location $\ell$. By construction, the generator has time-reversal symmetry, giving $I_{\ell;L}^{\mathrm{BP}} = I_{L-\ell;L}^{\mathrm{BP}}$.

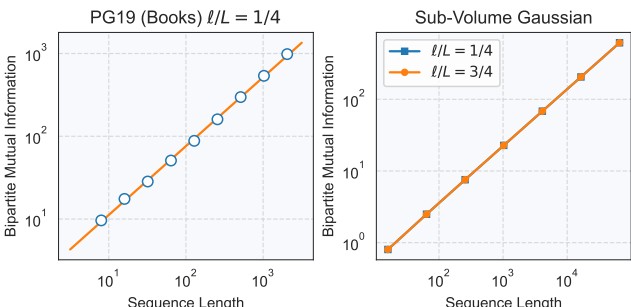

*Figure C.5.* Bipartite mutual information with asymmetric split. Both natural language and L-CUBE exhibit sub-volume power-law scaling with the same exponent regardless of split location, though the absolute magnitude depends on $\ell$. Natural language data are taken from Chen et al. (2025).

Fig. C.5 shows that asymmetric splits preserve the power-law scaling exponent while reducing the absolute magnitude.

### C.VII. Two-Point Mutual Information

Beyond matching bipartite mutual information scaling, L-CUBE also reproduces the power-law decay of two-point mutual information observed in natural language. The two-point mutual information $I(W_i; W_j)$ measures the statistical dependence between individual positions $i$ and $j$, and has been documented to decay as $I(W_i; W_j) \sim |i - j|^{-\alpha}$ with distance in natural language corpora (Chen et al., 2025; Ebeling & Pöschel, 2002).

This power-law decay reflects the hierarchical structure of language: nearby positions (within words or phrases) are strongly correlated, while distant positions (across sentences or paragraphs) retain weaker but non-negligible correlations through shared topics and entities. The power-law form indicates that correlations decay gradually rather than exponentially, characteristic of systems with multi-scale structure.

L-CUBE exhibits similar power-law decay through its hierarchical construction combined with local correlation. The reuse mechanism creates long-range dependencies through coarse scales, while local correlation ensures smooth decay at short

distances. Fig. C.6 shows that the sub-volume configuration matches the power-law exponent observed in natural language, confirming that L-CUBE captures both the global scaling properties (via $I^{\mathrm{BP}}$) and local correlation structure (via two-point MI) of linguistic dependencies.

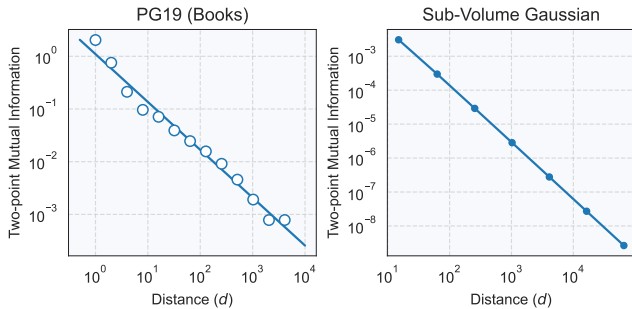

*Figure C.6.* Two-point mutual information $I(W_i; W_j)$ versus distance $|i-j|$ for natural language and L-CUBE sub-volume configuration. Both exhibit power-law decay with similar exponents, demonstrating that L-CUBE reproduces the correlation structure of natural language at multiple scales. Natural language results are taken from Chen et al. (2025).

## D. Connection to MERA

The Multi-scale Entanglement Renormalization Ansatz (MERA) (Vidal, 2007; 2008) is a tensor network architecture from quantum many-body physics designed to efficiently represent ground states of critical quantum systems in one dimension. MERA organizes a quantum state through a hierarchical structure of unitary or isometric tensors arranged in layers, where each layer consists of disentangling unitaries followed by isometric coarse-graining maps that reduce the system size by a constant factor (typically 2 or 3).

For one-dimensional critical systems described by conformal field theories, MERA achieves efficient representation by exploiting the fact that entanglement entropy grows only logarithmically with subsystem size: $S(L) \sim c \log L$, where $c$ is the central charge. This logarithmic scaling is intermediate between area-law systems (constant entanglement, $S = O(1)$) and generic states (volume-law, $S \sim L$). The isometric structure of MERA ensures that information is preserved through the hierarchy while the disentanglers reduce short-range entanglement at each scale.

L-CUBE can be viewed as a classical, probabilistic analog of MERA with important generalizations. The correspondence is:

| MERA (Quantum) | L-CUBE (Classical) |
| --- | --- |
| Quantum state $|\psi\rangle$ | Probability distribution $p(\boldsymbol{w})$ |
| Entanglement entropy $S$ | Mutual information $I$ |
| Isometric tensors | Block correlators $\mathcal{M}^{(d)}$ |
| Disentanglers | Local correlators |
| Bond dimension | Reuse parameter $m$ |
| Logarithmic entanglement | Constant/Logarithmic/(sub-)volume MI |

*Table D.1.* Correspondence between MERA and L-CUBE.

The key structural similarities are: (i) hierarchical organization across multiple scales, (ii) local mixing operations at each scale (isometries/disentanglers in MERA, block correlators $\mathcal{M}^{(d)}$ and local correlation in L-CUBE), and (iii) information flowing through the hierarchy via controlled pathways (bond dimension in MERA, reuse parameter $m$ in L-CUBE).

However, L-CUBE extends beyond standard MERA in several important ways. While MERA for 1D critical systems exhibits logarithmic entanglement scaling, achieving higher entanglement scaling would require increasing the bond dimension, leading to exponential growth in computational cost with system size. In contrast, L-CUBE supports arbitrary power-law scaling $I^{\mathrm{BP}} \sim L^\beta$ for any $\beta \in [0, 1]$ by varying the reuse parameter $m$, while maintaining $O(L)$ computational complexity for fixed $n, m$ with $m < n$ (and $O(L \log L)$ in the volume-law case $m = n$). When $m = 1$ with local correlation enabled, L-CUBE recovers logarithmic scaling analogous to MERA. When $1 < m < n$, L-CUBE exhibits sub-volume

power-law scaling with exponent $\beta = \log m / \log n$, a regime not naturally accessible in standard MERA without exponential cost. When $m = n$, L-CUBE achieves volume-law scaling. This flexibility allows L-CUBE to match the empirically observed sub-volume scaling in natural language, where $\beta \approx 0.7$-$0.8$. L-CUBE also operates on continuous Gaussian variables rather than quantum states, making it directly applicable to classical sequence modeling.

The connection to MERA provides theoretical grounding: both frameworks demonstrate that hierarchical organization with controlled information flow naturally produces scaling regimes beyond local (constant/area-law) dependencies. This universality across quantum and classical settings suggests that the scaling behavior is a fundamental property of hierarchical structure rather than specific to either domain.

# E. Experiment Details

## E.I. Continuous Adaptation

Language models are adapted to L-CUBE by replacing discrete token embeddings and softmax output layers with continuous input and Gaussian output parameterizations, without any tokenization or quantization of the Gaussian values. This adaptation preserves the core architectural mechanisms being evaluated (attention mechanisms, recurrent state updates, positional encoding) while enabling direct comparison against ground-truth conditionals provided by the generator. Sufficiently dense quantization of the Gaussian values would preserve mutual information scaling, but would forfeit the efficient exact sampling and inference that the continuous formulation supports.

The token embedding layer is replaced by a continuous adapter $f_{\text{in}} : \mathbb{R}^d \to \mathbb{R}^{d_{\text{model}}}$ that maps $d$-dimensional continuous inputs to the model's hidden dimension. At each position $t$, we use $d$ parallel independently generated Gaussian values from the hierarchical generator. These parallel sequences are independent instantiations of the same generative process, which multiplies the total mutual information by a constant factor $d$ while preserving the scaling behavior with sequence length. We typically use $d = 64$ to increase the absolute magnitude of information and stress-test model capacity. We use a multi-layer feedforward network with nonlinear activations to provide sufficient expressiveness for the continuous-to-vector mapping.

The output layer is replaced by a Gaussian parameterization head $f_{\text{out}} : \mathbb{R}^{d_{\text{model}}} \to \mathbb{R}^{2d}$ that predicts conditional mean and standard deviation for each of the $d$ dimensions:

$$(\hat{\mu}_t, \hat{\sigma}_t) = f_{\text{out}}(\mathbf{h}_t), \tag{E.1}$$

where $\mathbf{h}_t$ is the final-layer hidden state at position $t$, $\hat{\mu}_t, \hat{\sigma}_t \in \mathbb{R}^d$, and $\hat{\sigma}_t$ is constrained positive via a softplus activation. The model thus defines a conditional distribution $q(w_t \mid w_{1:t-1}) = \mathcal{N}(\hat{\mu}_t, \text{diag}(\hat{\sigma}_t^2))$ where the $d$ dimensions are conditionally independent given the history.

Models are trained by minimizing the per-position, per-dimension KL divergence between the true conditional $p(w_t \mid w_{1:t-1}) = \mathcal{N}(\mu_t, \text{diag}(\sigma_t^2))$ provided by the generator and the predicted conditional:

$$\mathcal{L} = \frac{1}{Ld} \sum_{t=1}^{L} D_{\text{KL}}(p(w_t \mid w_{1:t-1}) \| q(w_t \mid w_{1:t-1})), \tag{E.2}$$

where the KL divergence is over the full $d$-dimensional vector $w_t$. For diagonal Gaussian conditionals, this becomes

$$\mathcal{L} = \frac{1}{Ld} \sum_{t=1}^{L} \sum_{i=1}^{d} \left[ \log \frac{\hat{\sigma}_{t,i}}{\sigma_{t,i}} + \frac{\sigma_{t,i}^2 + (\mu_{t,i} - \hat{\mu}_{t,i})^2}{2\hat{\sigma}_{t,i}^2} - \frac{1}{2} \right]. \tag{E.3}$$

During evaluation, we use the trained conditionals $q(w_t \mid w_{1:t-1})$ and ground-truth conditionals $p(w_t \mid w_{1:t-1})$ to compute the utilization metric defined in Sec. 3.3. All dropout layers are disabled during both training and evaluation to isolate architectural capacity from regularization effects.

## E.II. Architecture Specifications

We evaluate models across multiple families and sizes, all adapted for continuous input/output as described above. Sequence lengths indicate the total sequence length $L$; the corresponding context length used for evaluation is $3L/4$.

**GPT-2 (Transformers):** Learned positional embeddings, standard attention.

- 64M: 12 layers, 512 hidden. Lengths: 64, 256, 1024, 4096, 16384

- 124M: 12 layers, 768 hidden. Lengths: 64, 256, 1024, 4096, 16384

- 355M: 24 layers, 1024 hidden. Lengths: 64, 256, 1024, 4096, 16384

- 774M: 36 layers, 1280 hidden. Lengths: 64, 256, 1024, 4096

**GPT-NeoX (Transformers):** RoPE, parallel attention/FFN.

- 70M: 6 layers, 512 hidden. Lengths: 64, 256, 1024, 4096, 16384

- 162M: 12 layers, 768 hidden. Lengths: 64, 256, 1024, 4096, 16384

- 405M: 24 layers, 1024 hidden. Lengths: 64, 256, 1024, 4096, 16384

- 1.01B: 16 layers, 2048 hidden. Lengths: 64, 256, 1024, 4096

**Qwen3 (Transformers):** RoPE, SwiGLU, RMSNorm.

- 110M: 8 layers, 512 hidden. Lengths: 64, 256, 1024, 4096, 16384

- 224M: 12 layers, 768 hidden. Lengths: 64, 256, 1024, 4096

- 535M: 24 layers, 1024 hidden. Lengths: 64, 256, 1024, 4096

**Mamba (SSM):** State dimension 16, expansion factor 2.

- 129M: 24 layers, 768 hidden. Lengths: 64, 256, 1024, 4096, 16384

- 372M: 48 layers, 1024 hidden. Lengths: 64, 256, 1024, 4096, 16384

- 793M: 48 layers, 1536 hidden. Lengths: 64, 256, 1024, 4096, 16384

- 1.37B: 48 layers, 2048 hidden. Lengths: 64, 256, 1024, 4096, 16384

**Mamba2 (SSM):** State dimension 128, multi-head structure.

- 130M: 24 layers, 768 hidden. Lengths: 64, 256, 1024, 4096, 16384

- 371M: 48 layers, 1024 hidden. Lengths: 64, 256, 1024, 4096, 16384

- 785M: 48 layers, 1536 hidden. Lengths: 64, 256, 1024, 4096, 16384

- 1.35B: 48 layers, 2048 hidden. Lengths: 64, 256, 1024, 4096, 16384

**RWKV-7 (SSM):** Improved time-mixing with squared ReLU.

- 72M: 6 layers, 512 hidden. Lengths: 64, 256, 1024, 4096, 16384

- 164M: 12 layers, 768 hidden. Lengths: 64, 256, 1024, 4096, 16384

- 401M: 24 layers, 1024 hidden. Lengths: 64, 256, 1024, 4096

- 811M: 24 layers, 1536 hidden. Lengths: 64, 256, 1024, 4096

- 1.36B: 24 layers, 2048 hidden. Lengths: 64, 256, 1024, 4096

**LLA (Log-Linear Attention):** Polynomial feature map (degree 2), feature dimension 256.

- 40M: 12 layers, 512 hidden. Lengths: 64, 256, 1024, 4096, 16384

- 177M: 24 layers, 768 hidden. Lengths: 64, 256, 1024, 4096, 16384

- 621M: 48 layers, 1024 hidden. Lengths: 64, 256, 1024, 4096, 16384

- 1.39B: 48 layers, 1536 hidden. Lengths: 64, 256, 1024, 4096

**Mistral (Sliding Window Attention):** 430M parameters, 24 layers, 1024 hidden, GQA with 2 KV heads.

- Window 64: Lengths: 64, 256, 1024, 4096, 16384

- Window 256: Lengths: 64, 256, 1024, 4096, 16384

- Window 4096: Lengths: 64, 256, 1024, 4096, 16384

### E.III. Training Hyperparameters

All models are trained using the AdamW optimizer (Kingma & Ba, 2015) with the following hyperparameters unless otherwise specified:

*Table E.1.* Common training hyperparameters across all experiments.

| Hyperparameter | Value |
|---|---|
| Optimizer | AdamW |
| Learning rate | $5 \times 10^{-5}$ |
| Weight decay | 0.01 |
| LR warmup steps | 2,000 |
| LR warmup initial factor | $10^{-7}$ |
| LR scheduler | Linear warmup + Cosine annealing |
| Gradient norm clip | 2.0 (L2 norm) |
| Gradient value clip | 0.2 |
| Precision | Float32 (BFloat16 is used if Float32 is not supported) |
| Maximum training steps | 500,000 |
| Effective batch size | 4 sequences |

### E.IV. Computational Resources

**Hardware configuration:** Experiments were primarily conducted on NVIDIA GH200 Grace Hopper Superchip nodes.

**Total compute:** Approximately 80,000 GPU hours across all experiments.

### E.V. Reproducibility

Code and data are available at https://github.com/LSquaredM/L-CUBE. A public challenge based on the benchmark will also be made available.

## F. Complete Experimental Results

This section provides comprehensive experimental results for all model sizes and architectures evaluated in the main text. We begin with detailed utilization curves extending Fig. 3, then analyze the KL residuals and raw conditional KL divergences that compose utilization. Per-position analysis reveals where in sequences different architectures struggle, and computational complexity analysis extends the scaling observations from the main text.

## F.I. Long-Context Utilization Across All Model Sizes

Figures F.1-F.4 show utilization $U(X;Y)$ versus sequence length for all architectures and model sizes, extending the representative results from Fig. 3 in the main text. The black dashed line indicates ground-truth bipartite mutual information $I^{\mathrm{BP}}_{3L/4;L}$ provided by the generator.

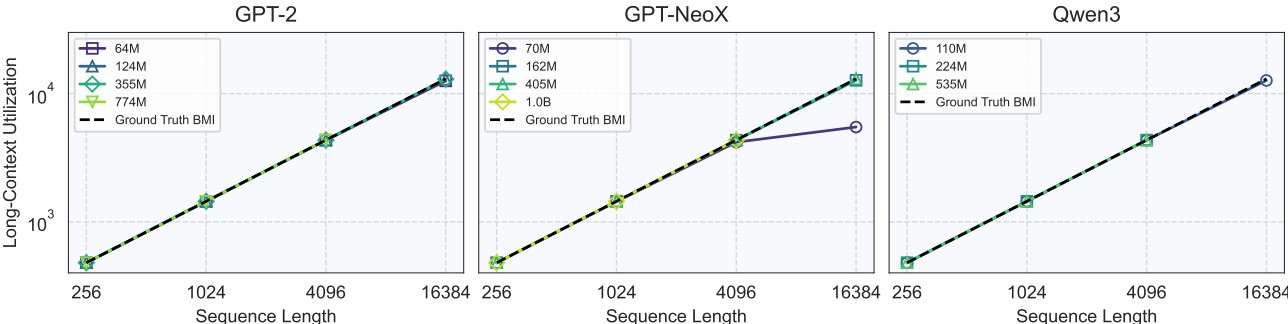

*Figure F.1.* Long-context utilization for transformers across all model sizes (64M to 774M parameters). Most models track ground truth across tested lengths, with larger models achieving slightly better utilization. The GPT-NeoX 70M model is the main exception at the longest lengths. Even the 64M GPT-2 model maintains high utilization, confirming that transformers generally have sufficient architectural capacity for sub-volume scaling.

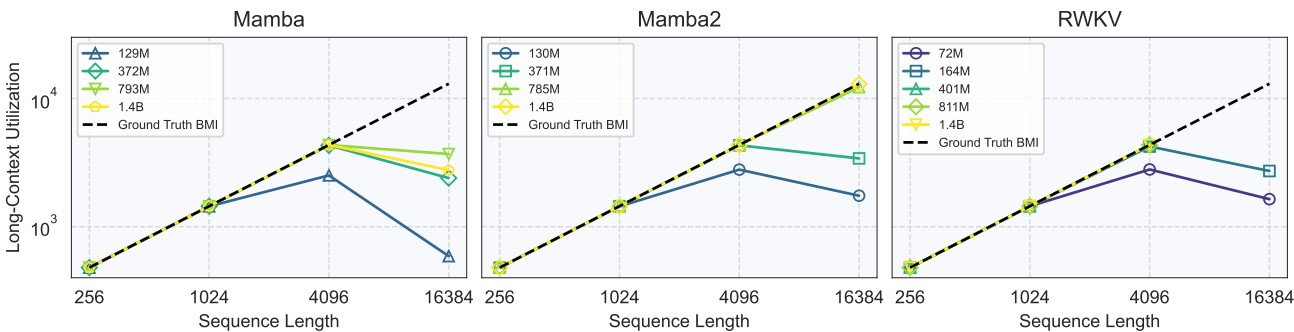

*Figure F.2.* Long-context utilization for SSMs across all model sizes (129M to 1.37B parameters), including Mamba, Mamba2, and RWKV-7. All SSMs eventually plateau or decline as predicted by fixed state capacity, but larger models maintain high utilization to longer lengths. RWKV-7 performs best within each size class, followed by Mamba2, then Mamba, reflecting architectural differences in effective state capacity.

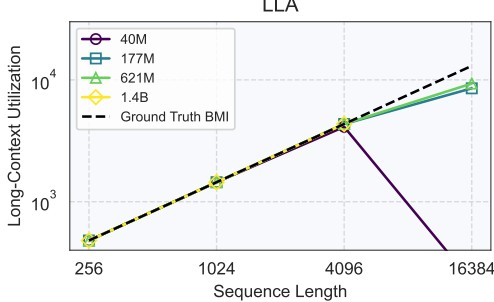

*Figure F.3.* Long-context utilization for LLA across all model sizes (40M to 1.39B parameters). Performance improves with model size but all variants show degradation at long lengths, consistent with logarithmic capacity growth being insufficient for sub-volume scaling.

The complete results confirm the patterns observed in the main text. Transformers more successfully track $I^{\mathrm{BP}}$ across the tested range. The smallest 64M transformer maintains utilization comparable to models 10× larger, demonstrating that architectural capacity, not parameter count alone, determines long-context capability for transformers when models are appropriately sized. The exception is GPT-NeoX 70M, which shows degradation at long lengths, likely due to interactions between rotary positional embeddings and very small model size.

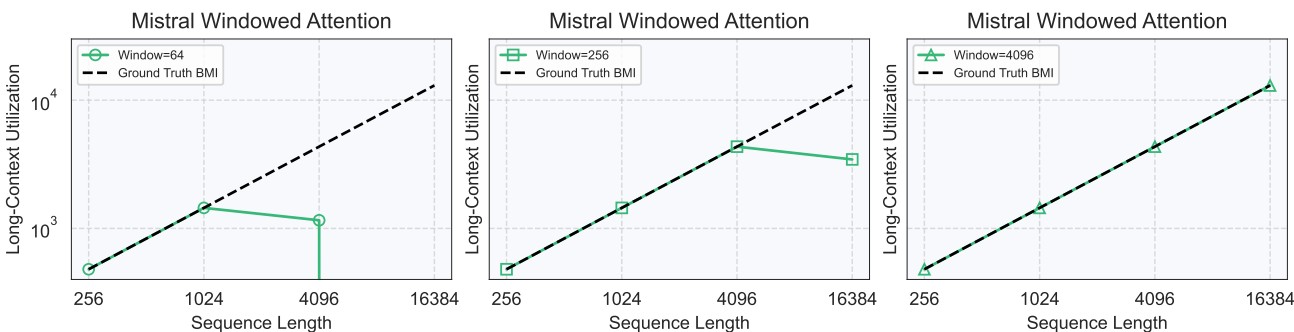

*Figure F.4.* Long-context utilization for Mistral (430M parameters) with different window sizes (64, 256, 4096 tokens). Each configuration tracks ground truth while context remains within its context window, then shows sharp degradation when sequence length exceeds the context window boundary. Larger windows maintain utilization to correspondingly longer sequence lengths.

For SSMs, RWKV-7 maintains utilization longest, followed by Mamba2, then Mamba. This ordering reflects architectural differences in effective state capacity. Within each architecture, larger models delay the onset of degradation but do not prevent it. The slope of utilization decline is similar across sizes, with larger models simply shifting the critical length rightward. This confirms that the failure mode is architectural rather than a matter of parameter count.

LLA shows intermediate behavior, performing better than SSMs due to its logarithmic state growth but still failing to match transformers at long lengths. Mistral's performance is entirely determined by window size: each window configuration behaves identically to a full-attention transformer within the context window size and fails abruptly beyond it, providing clean empirical validation of the context window size as a hard architectural constraint.

## F.II. Relative Residual Analysis

The relative residual $\rho(X; Y) = \Delta_{\mathrm{KL}}(X; Y)/G(X; Y)$ provides a normalized measure of utilization failure, representing the fraction of available information that models fail to capture (Eq. 15). Figures F.5-F.8 show the relative residual $\rho(X; Y)$ versus sequence length.

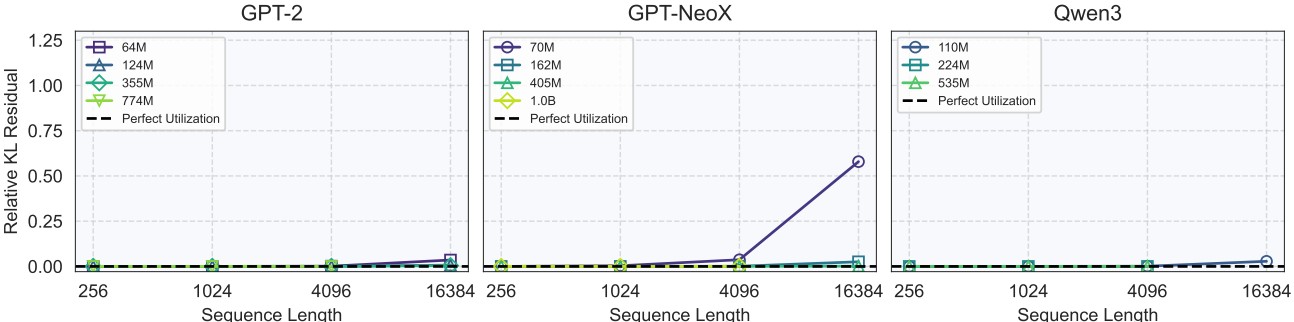

*Figure F.5.* Relative residual for transformers across all model sizes.

## F.III. KL Divergence Measurements

While utilization $U(X; Y)$ measures realized gain from context, the underlying KL divergences $D_{\mathrm{KL}}(p(y \mid x)\|q(y \mid x))$ reveal the absolute quality of learned distribution. Figures F.9-F.12 show KL divergence versus sequence length for all architectures and model sizes.

## F.IV. Per-Position Analysis

Figures F.13-F.16 show conditional KL divergence as a function of token position for various sequence lengths. These per-position curves reveal where in the sequence different architectures struggle.

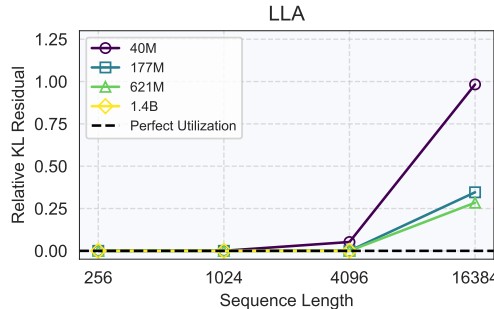

*Figure F.6.* Relative residual for SSMs across all model sizes.

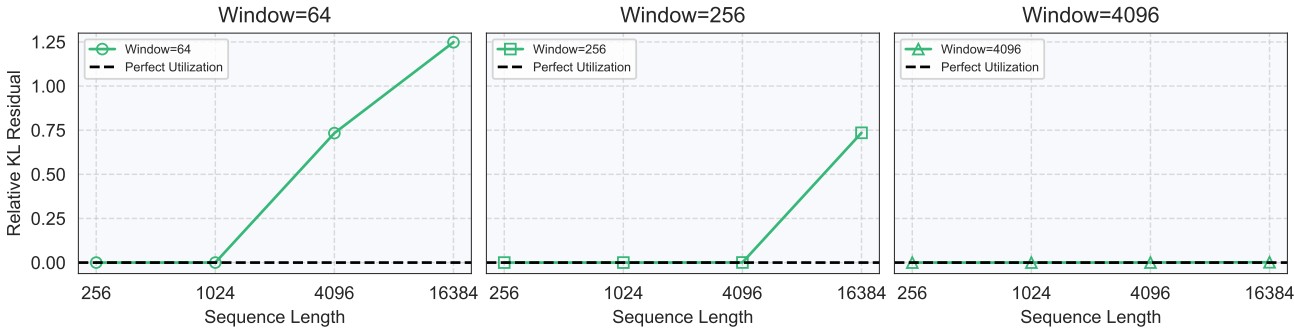

*Figure F.7.* Relative residual for LLA across all model sizes.

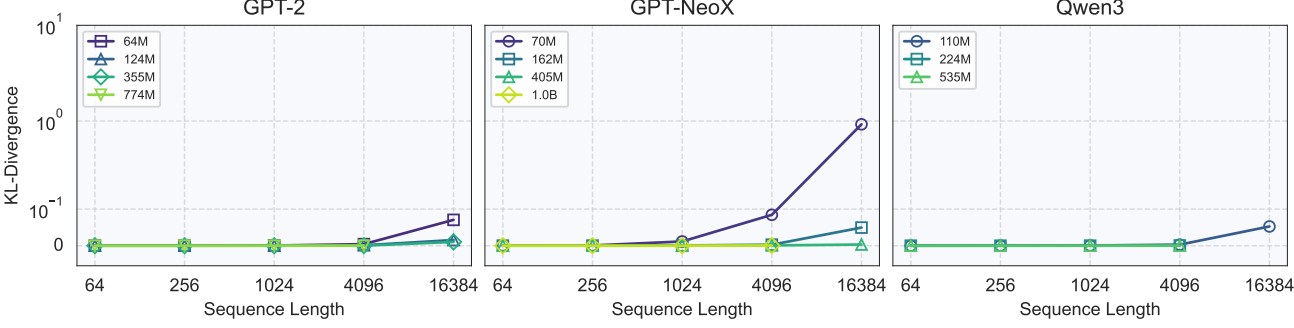

*Figure F.8.* Relative residual for Mistral with different window sizes.

*Figure F.9.* KL divergence for transformers across all model sizes. Transformers maintain relatively low KL divergence across all tested sequence lengths, with larger models achieving lower absolute values. The consistent performance across lengths confirms transformers' capacity for long-range dependencies. GPT-NeoX 70M shows elevated KL at long lengths, suggesting interaction between rotary positional embeddings and minimal model capacity.

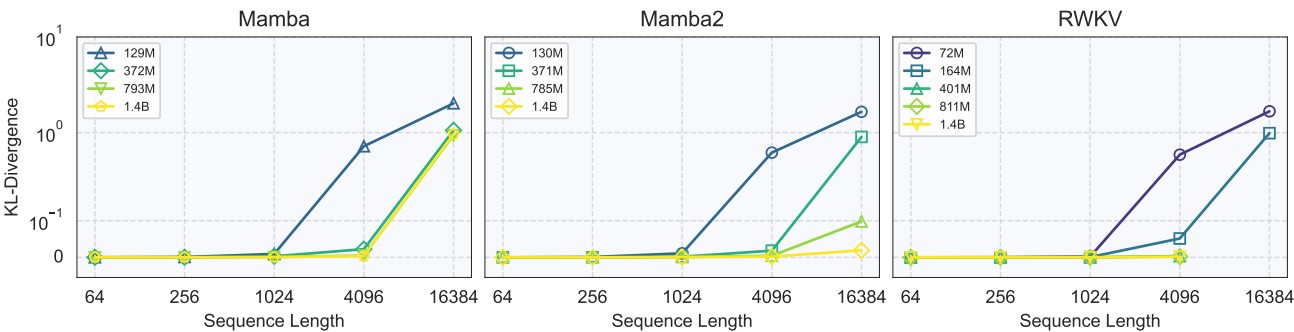

*Figure F.10.* KL divergence for SSMs across all model sizes. SSMs show increasing KL divergence as sequence length grows, with the divergence onset occurring at longer lengths for larger models. This growth reflects the capacity bottleneck predicted by $L^2M$ theory: as more context must be compressed into fixed-size state, conditional quality degrades.

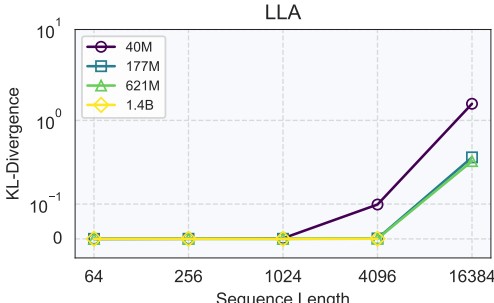

*Figure F.11.* KL divergence for LLA across all model sizes. LLA exhibits intermediate behavior between transformers and SSMs, with KL divergence increasing more slowly than SSMs but faster than transformers.

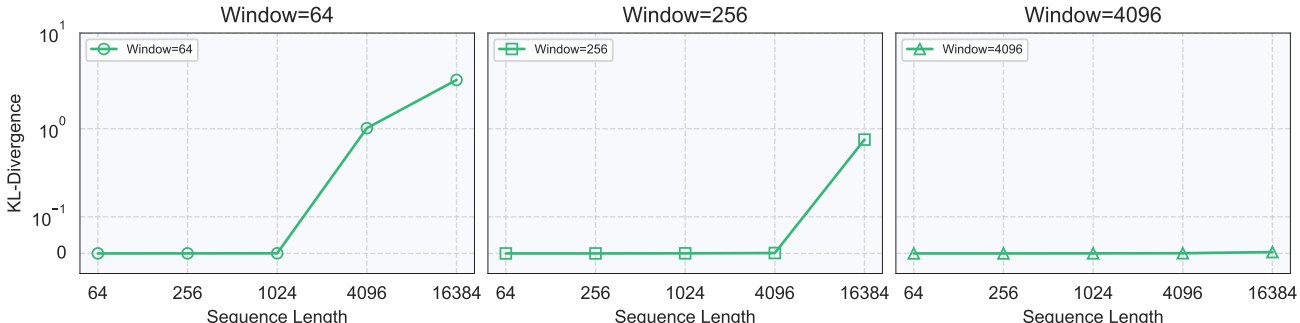

*Figure F.12.* KL divergence for Mistral with different window sizes. KL divergence remains low while context stays within the window, then increases sharply when sequence length exceeds the window size, demonstrating the hard limitation of finite attention span.

### F.V. Computational Complexity Analysis

Fig. F.17 extends Fig. 5(b) from the main text by showing how the compute-utilization relationship varies across different context lengths within each model, not just the maximum utilized context length.

### F.VI. Utilization During Training

Figs. F.18–F.23 show long-context utilization at intermediate training checkpoints (50k, 100k, 150k, 250k, 350k, and 500k steps). Architectural rankings are consistent throughout training: transformers track the ground truth early, SSMs show declining utilization at long sequences from early stages, and LLA occupies an intermediate position. Windowed attention with $w = 64$ shows declining utilization at longer sequences as training progresses; because utilization measures long-range dependency capture, the window size limitation forces the model to prioritize local patterns, which can reduce utilization even as local prediction quality improves.

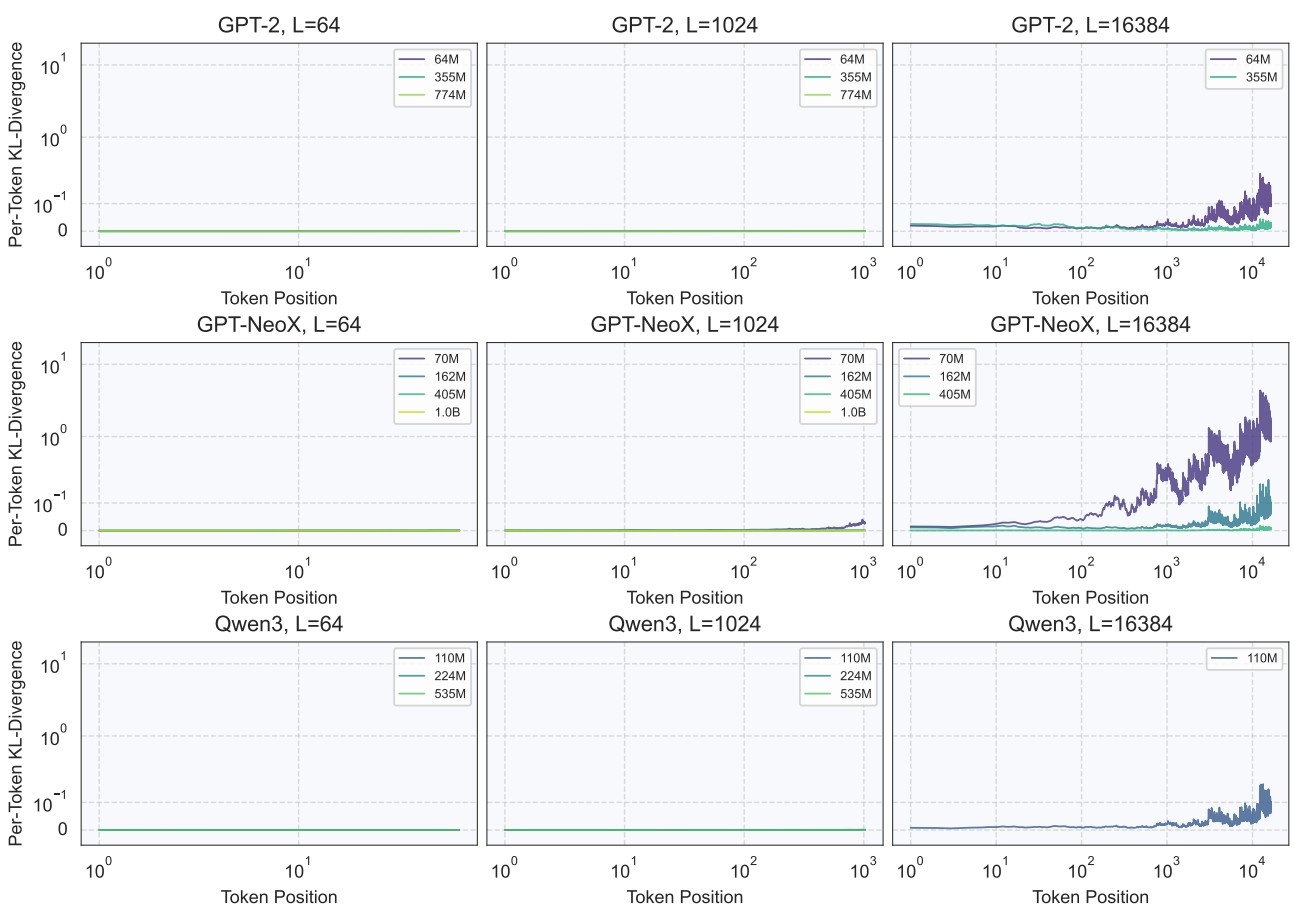

*Figure F.13.* Per-position conditional KL divergence for transformers.

## F.VII. Training Convergence

Figs. F.24–F.26 show evaluation loss during training at sequence lengths 64, 1024, and 16384. All architectures converge well before the 500,000-step training limit, confirming that the utilization comparisons in the main text reflect converged performance rather than optimization differences.

## F.VIII. Seed Variance

Fig. F.27 shows evaluation loss during training under different random seeds at sequence length 4096 for GPT-NeoX (162M) and LLA (177M). Training curves are consistent across seeds.

## G. Logarithmic Scaling Regime

The main-text results use sub-volume MI scaling ($\beta \approx 0.79$). Figs. G.1 and G.2 show utilization and raw KL divergence in the logarithmic regime ($m = 1$ with local correlation), where $I^{\mathrm{BP}}$ grows as $O(\log L)$. GPT-NeoX tracks the ground truth at shorter lengths but utilization declines at the longest tested sequence. Mamba's fixed state is insufficient even for slowly growing MI. LLA tracks the ground truth across all tested lengths. Despite GPT-NeoX's larger history state, LLA outperforms it in this regime, possibly because vanilla RoPE limits GPT-NeoX's effective capacity or because matching the data's MI scaling rate is more effective than exceeding it, which is not predicted by the $L^2M$ theory.

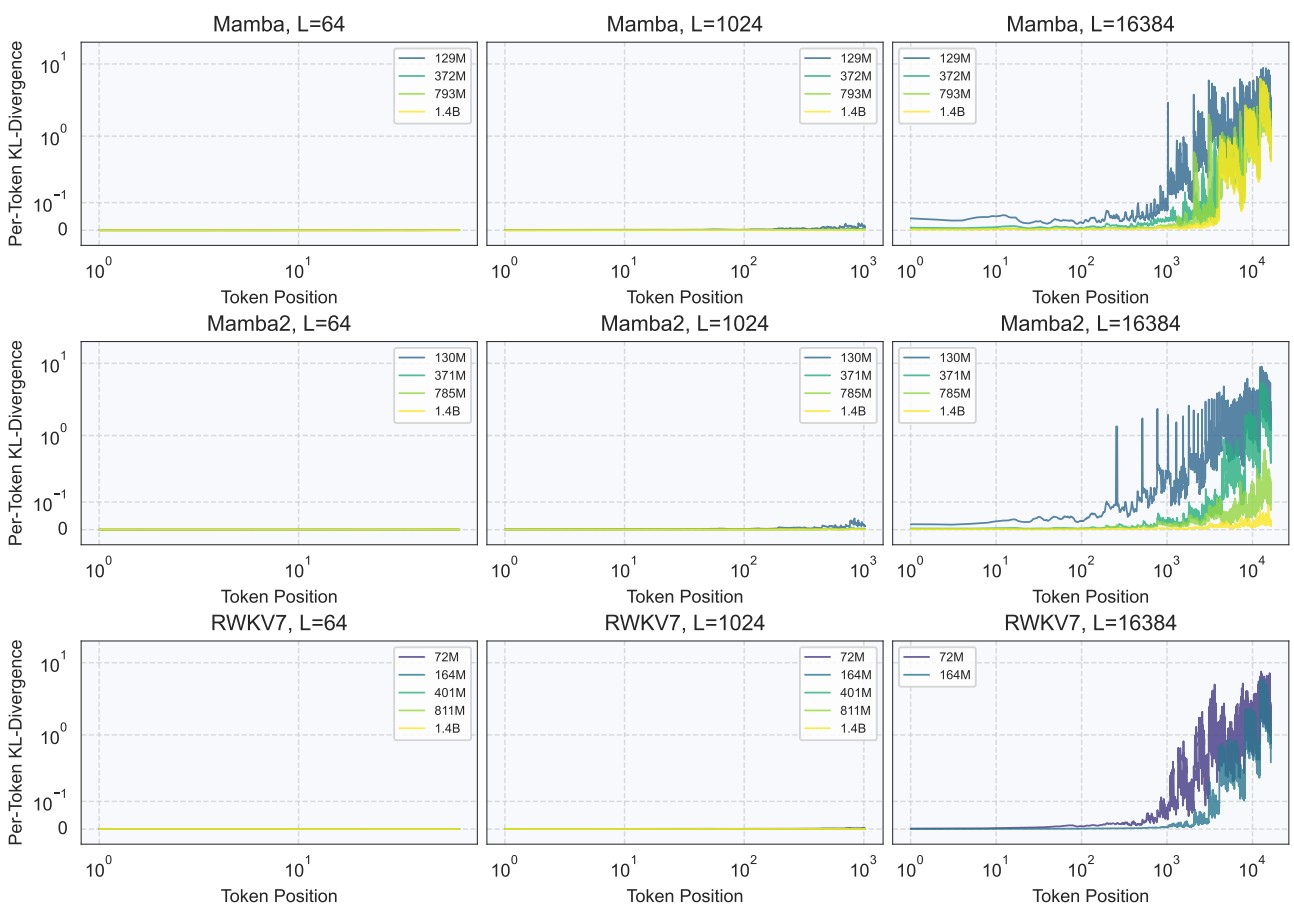

*Figure F.14.* Per-position conditional KL divergence for SSMs.

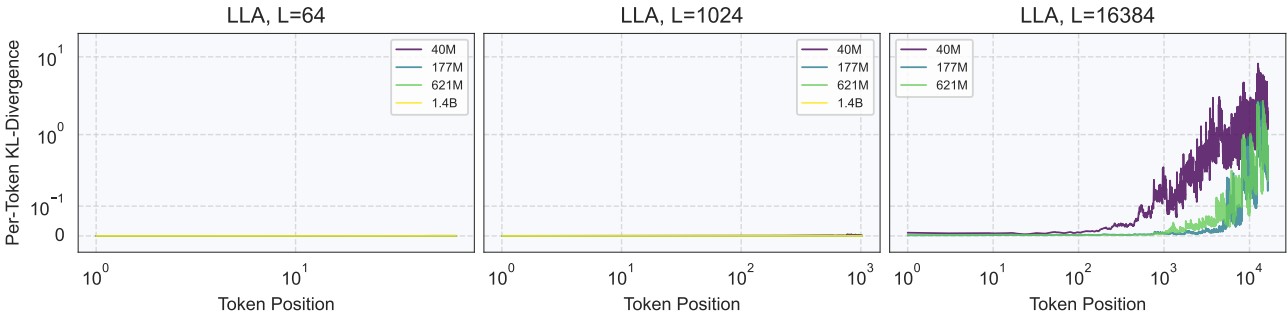

*Figure F.15.* Per-position conditional KL divergence for LLA.

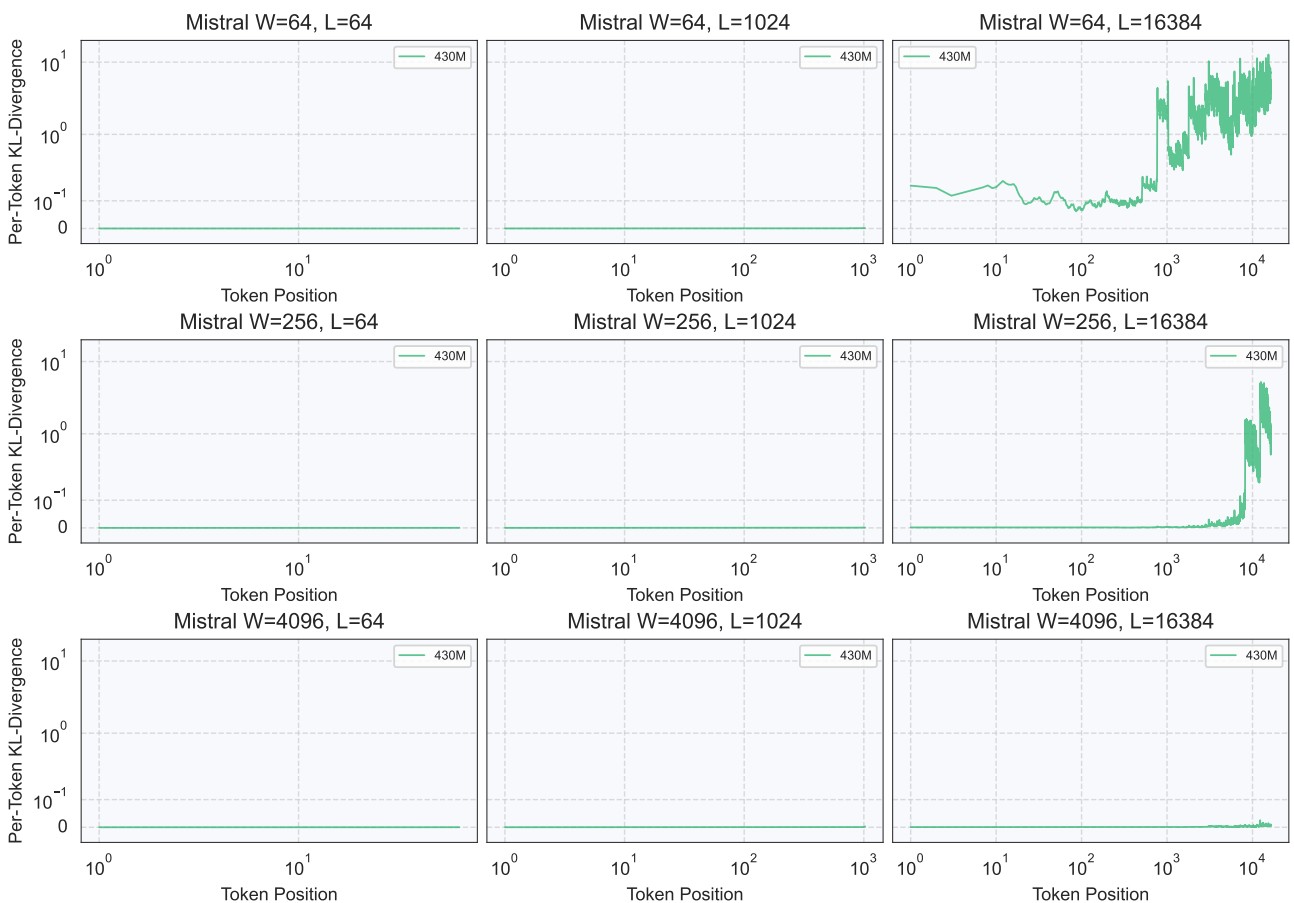

*Figure F.16.* Per-position conditional KL divergence for Mistral.

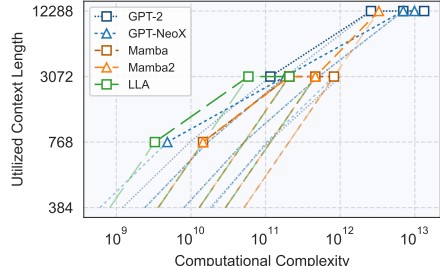

*Figure F.17.* Utilized context length versus theoretical computational complexity across individual model and context-length settings. This extends Fig. 5(b) by showing the compute-utilization relationship across tested lengths, rather than only each model's maximum utilized context length.

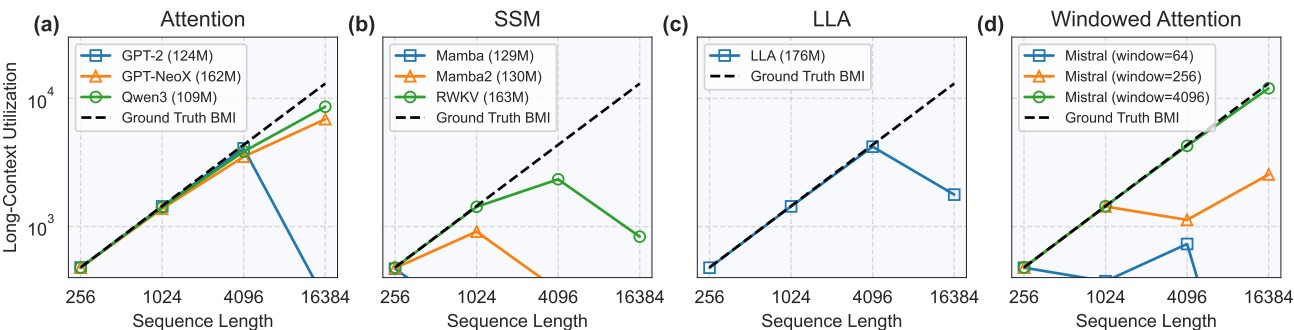

*Figure F.18.* Long-context utilization at 50,000 training steps for (a) transformers, (b) SSMs, (c) LLA, and (d) windowed attention.

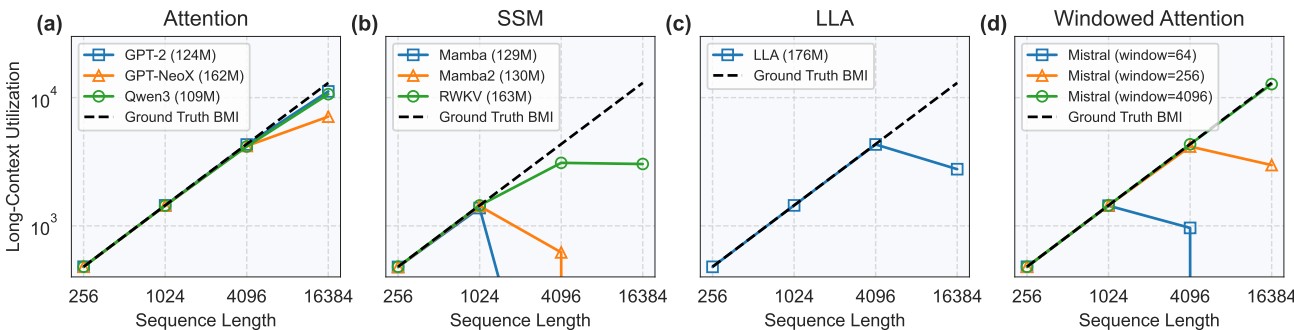

*Figure F.19.* Long-context utilization at 100,000 training steps for (a) transformers, (b) SSMs, (c) LLA, and (d) windowed attention.

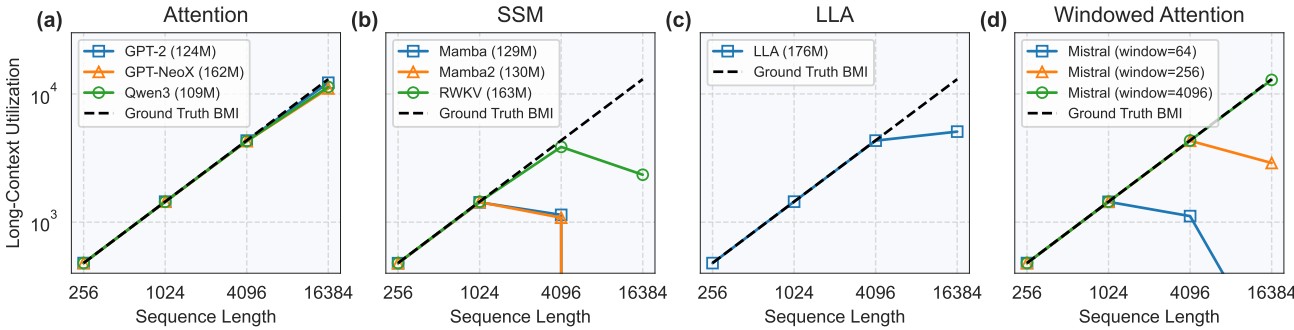

*Figure F.20.* Long-context utilization at 150,000 training steps for (a) transformers, (b) SSMs, (c) LLA, and (d) windowed attention.

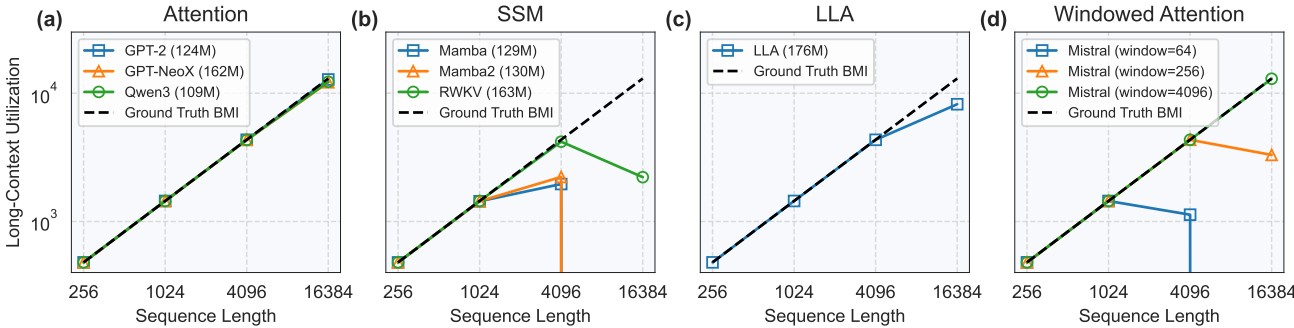

*Figure F.21.* Long-context utilization at 250,000 training steps for (a) transformers, (b) SSMs, (c) LLA, and (d) windowed attention.

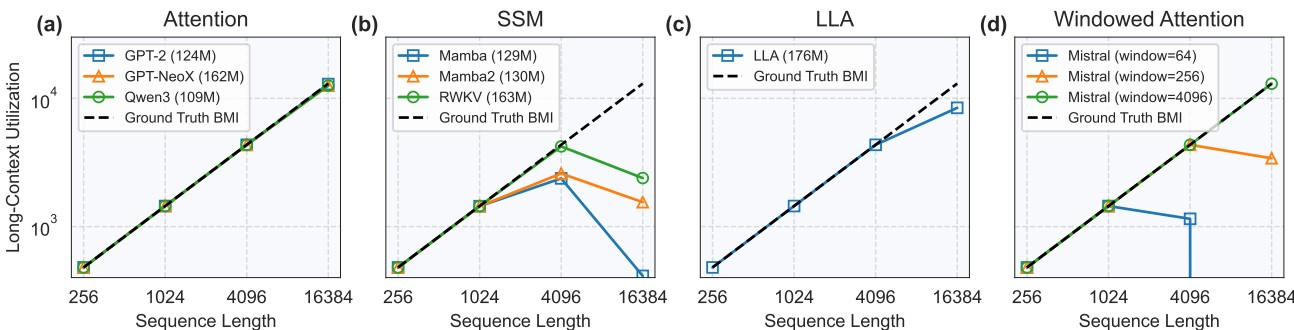

*Figure F.22.* Long-context utilization at 350,000 training steps for (a) transformers, (b) SSMs, (c) LLA, and (d) windowed attention.

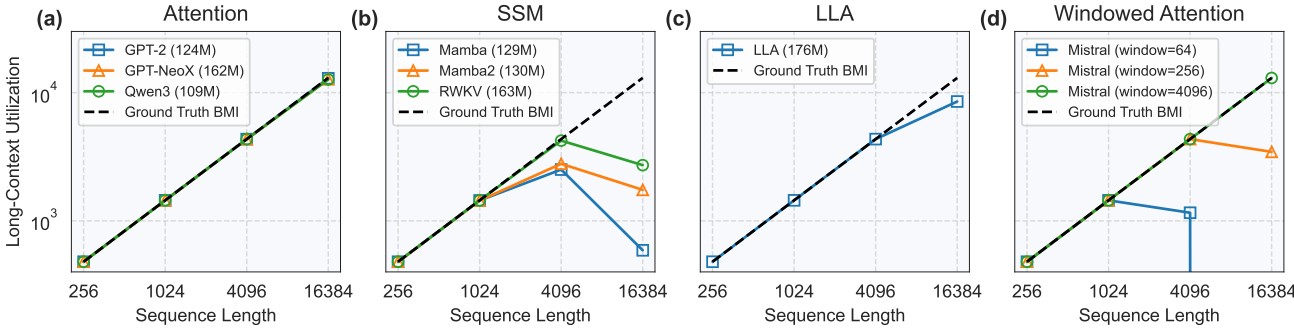

*Figure F.23.* Long-context utilization at 500,000 training steps for (a) transformers, (b) SSMs, (c) LLA, and (d) windowed attention.

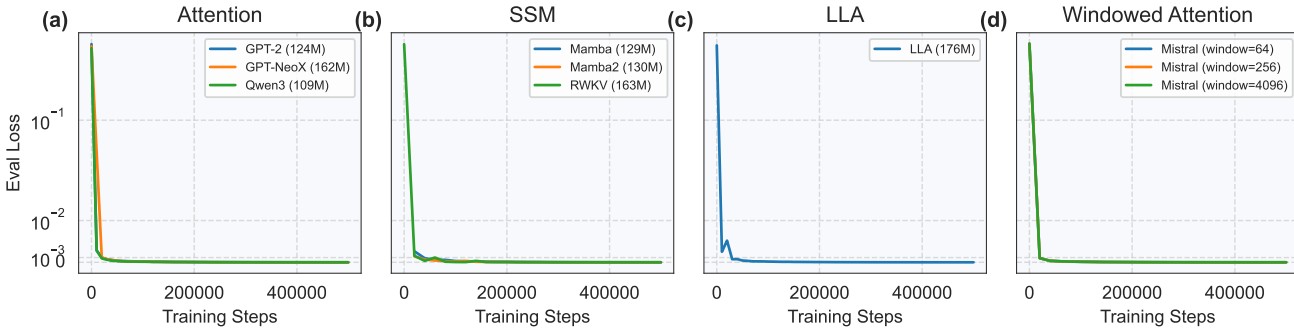

*Figure F.24.* Evaluation loss during training at sequence length 64 for (a) transformers, (b) SSMs, (c) LLA, and (d) windowed attention.

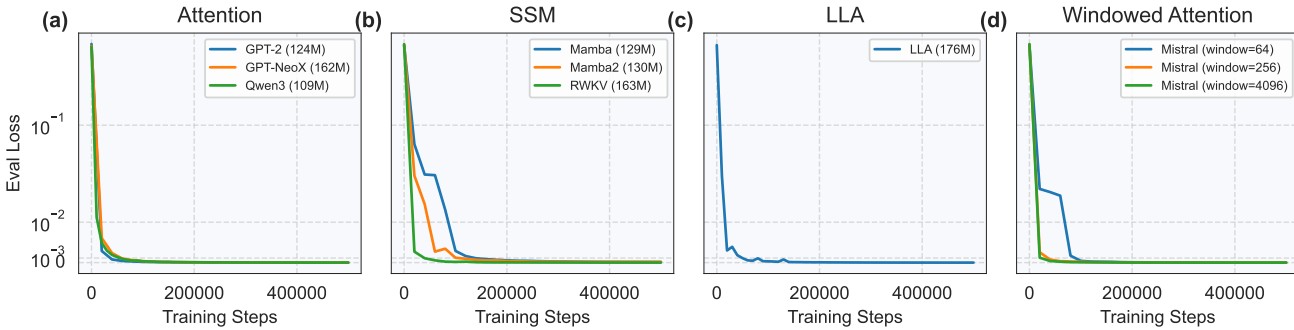

*Figure F.25.* Evaluation loss during training at sequence length 1024 for (a) transformers, (b) SSMs, (c) LLA, and (d) windowed attention.

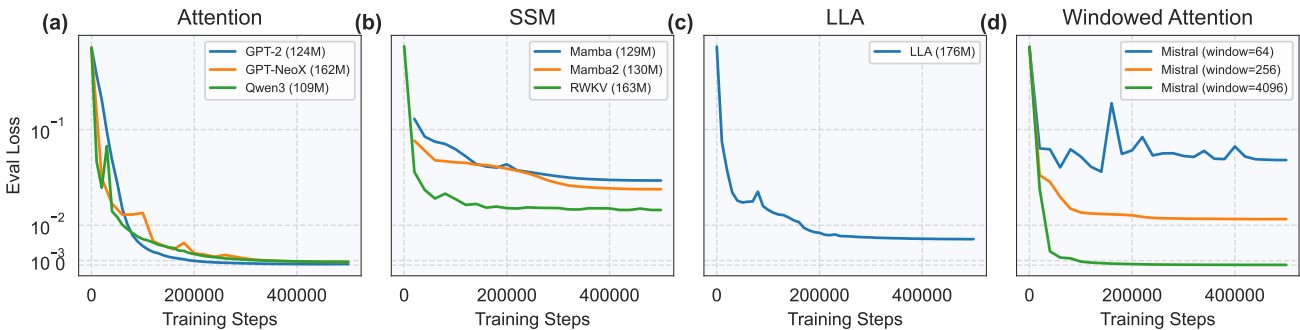

*Figure F.26.* Evaluation loss during training at sequence length 16384 for (a) transformers, (b) SSMs, (c) LLA, and (d) windowed attention.

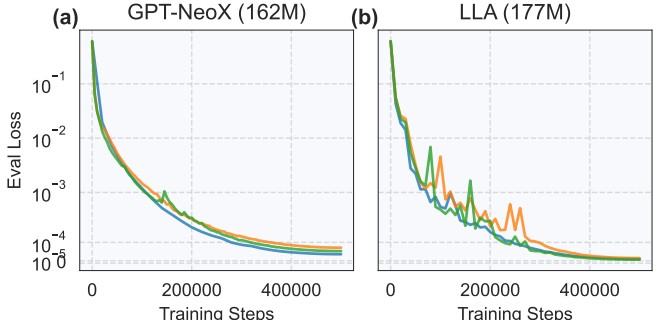

*Figure F.27.* Evaluation loss during training under different random seeds at sequence length 4096 for (a) GPT-NeoX (162M) and (b) LLA (177M).

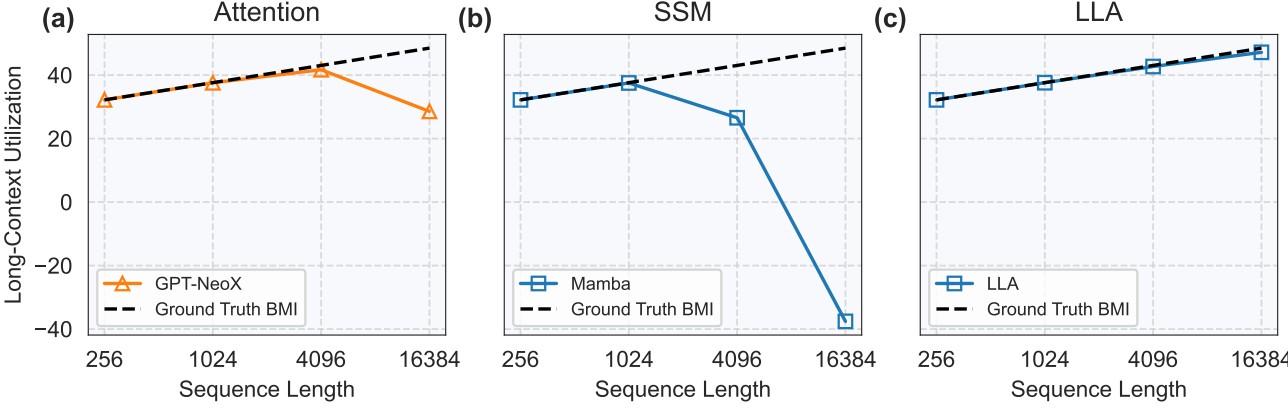

*Figure G.1.* Long-context utilization in the logarithmic MI scaling regime for (a) GPT-NeoX, (b) Mamba, and (c) LLA. The black dashed line indicates ground-truth bipartite mutual information $I^{\mathrm{BP}}_{3L/4;L}$.

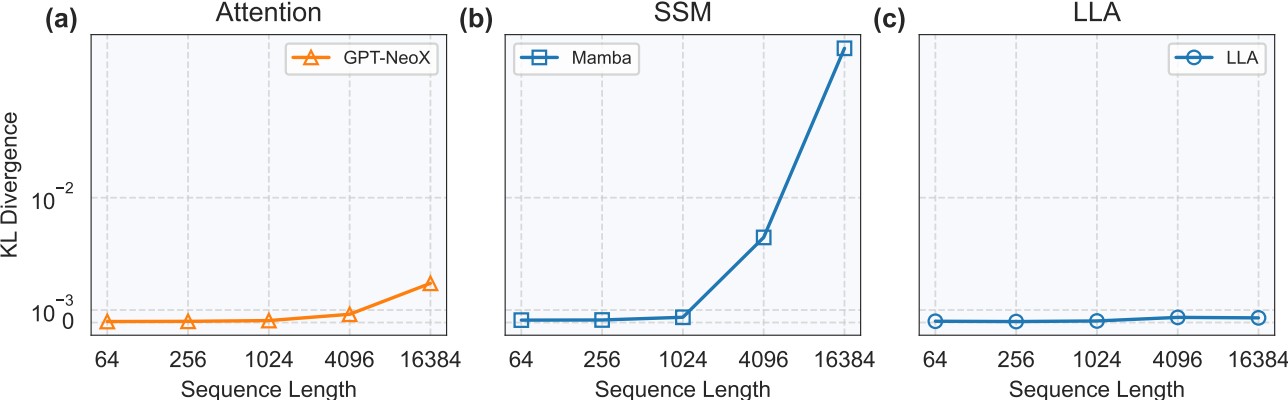

*Figure G.2.* Raw KL divergence in the logarithmic MI scaling regime for (a) GPT-NeoX, (b) Mamba, and (c) LLA.

# H. Pretraining Augmentation

We investigate whether augmenting pretraining with L-CUBE sequences improves length generalization when natural long-context data is limited. The central question is whether synthetic sequences with controlled hierarchical structure can help models trained primarily on short sequences generalize to longer contexts.

## H.I. Experimental Setup

We train Pythia-410M models (GPT-NeoX architecture) from scratch on the PG19 dataset. The training data consists of three components: (1) primarily short sequences (256 tokens), (2) a small fraction of long real sequences (sequences > 256 tokens), and (3) synthetic long sequences generated by L-CUBE. We vary the fraction of long real data between 0.1% and 1% of the total training mixture, while keeping the overall training distribution balanced between short and long contexts. Models are evaluated at longer sequence lengths (1024 and 4096 tokens) where they must extrapolate beyond the typical training length.

For these experiments, we discretize the continuous L-CUBE sequences to better mimic discrete tokens. The continuous Gaussian values are binned into discrete values, creating token-like sequences that can be processed by standard language model tokenization. We use 16 parallel independent sequences at each position, which increases the total information content while maintaining the hierarchical scaling structure. To restore translation invariance (which is broken by the hierarchical construction's specific boundary positions), we generate sequences 4× longer than needed and randomly extract continuous subsequences of the target length. We use variance parameters $[1.0, 0.1111, 0.1666, 0.1111]$ (normalized to sum $n = 4$), to produce correlation structure suitable for the discretized setting.

We compare models trained with L-CUBE augmentation against baseline models trained on the same mixture of short and long real data without synthetic augmentation. This isolates the effect of synthetic data on length generalization capability.

## H.II. Results

Fig. H.1 shows three experimental comparisons testing different aspects of synthetic data augmentation:

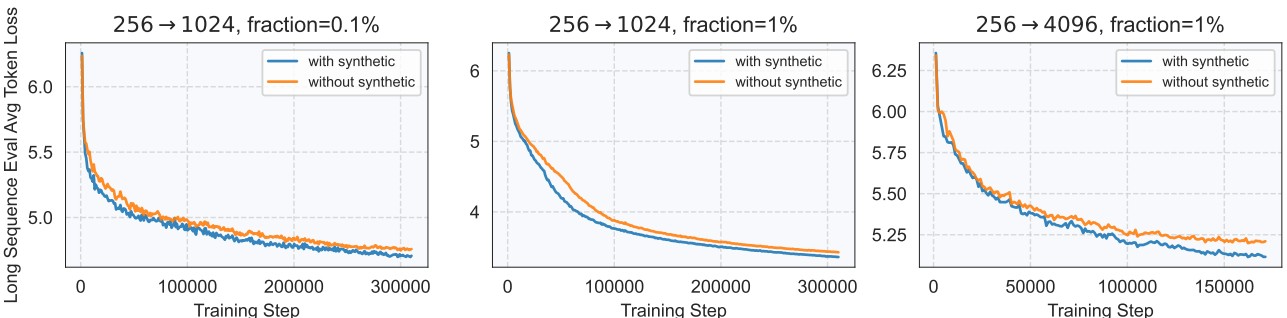

*Figure H.1.* Length generalization with and without L-CUBE augmentation for Pythia-410M (GPT-NeoX) trained on PG19. Training data consists of primarily short sequences (256 tokens) plus a small fraction of long real data and (for augmented models) synthetic long data. Left: 0.1% long real data, evaluated at 1024 tokens. Middle: 1% long real data (10× more), evaluated at 1024 tokens. Right: 1% long real data, evaluated at 4096 tokens (harder extrapolation). Synthetic augmentation provides the largest benefit when real long-context data is scarce and extrapolation is challenging.

These preliminary results suggest that L-CUBE augmentation is most beneficial when (1) real long-context data is scarce and (2) the target evaluation length requires substantial extrapolation beyond typical training length. The mechanism likely involves synthetic data teaching models to utilize hierarchical long-range dependencies without the confound of semantic complexity. However, these are early-stage results on a single model size and dataset. Comprehensive validation across model scales, architectural families, datasets, mixing strategies, and discretization approaches is needed before drawing strong conclusions about optimal training recipes.

# I. Practical Usage Guide

## I.I. When to Use L-CUBE

L-CUBE is designed for architectural diagnosis in the following scenarios:

- **Before committing to expensive real-data training:** Test whether an architecture has sufficient capacity for target sequence lengths.

- **When comparing architectural modifications:** Isolate the effect of design changes on long-context capacity.

- **To diagnose why a model fails at long contexts:** Determine whether failures stem from capacity limits or other factors.

## I.II. Interpreting Results

- **If utilization tracks $I^{\mathrm{BP}}$:** The architecture has sufficient capacity to capture hierarchical long-range dependencies at the tested lengths.

- **If utilization plateaus or declines:** The architecture has fundamental capacity limits at that sequence length, indicating that the history state dimension is insufficient.

- **Relative residual $\rho < 1\%$:** Indicates near-optimal capacity utilization; the model captures over 99% of available information.

## I.III. What L-CUBE Does Not Predict

By design, L-CUBE diagnoses whether an architecture *can* succeed at long contexts, not whether it *will* succeed after training on specific data. Success on L-CUBE is a necessary but not sufficient condition for real-world long-context performance. L-CUBE does not replace benchmarks that evaluate semantic understanding, reasoning, or task-specific capabilities.

# J. Limitations

L-CUBE measures architectural capacity for hierarchical long-range dependencies but does not evaluate semantic understanding, reasoning, in-context learning, or natural language statistics. An architecture that fails on L-CUBE will likely struggle with any long-context task, but success on L-CUBE does not guarantee success on semantic tasks. By design, it provides a necessary condition for long-context modeling, not a sufficient one. Our evaluation is autoregressive, although the generator itself is direction-free and supports bidirectional and alternative masking conventions, which we leave for future work.

Results on pretraining augmentation (Appx. H) represent an initial exploration; the computational cost of training at scale limits comprehensive validation across model sizes and training configurations, which we leave for future work. Similarly, the compute-scaling regularity (Fig. 5b) would benefit from validation on broader architecture sets. We evaluate representative architectures from major families but cannot exhaustively test every variant. Hybrid designs that interleave full-attention layers with sub-quadratic ones inherit attention's asymptotic compute and memory cost, so they should behave on L-CUBE much like pure-attention models of comparable size. The primary question L-CUBE asks is whether purely sub-quadratic architectures can represent the required mutual information growth, and we therefore benchmark such alternatives directly rather than tune attention/sub-quadratic mix ratios.

