# OpenReview forum: "L-CUBE: Isolating Long-Context Capacity from Knowledge with Controllable Mutual Information Scaling"
_ICML.cc/2026/Conference — ICML 2026 regular_

### Official Review · Reviewer_jC8j · 2026-03-10

**Soundness:** 3
**Presentation:** 3
**Significance:** 3
**Originality:** 4
**Overall Recommendation:** 4
**Confidence:** 1

**Summary:**

This paper proposes L-CUBE, a synthetic benchmark for evaluating long-context modeling ability while minimizing confounds from natural language semantics and pretrained knowledge. By constructing hierarchical Gaussian sequences with controllable long-range dependencies and known ground-truth distributions, the benchmark enables direct measurement of how well different architectures utilize long-context information. Experiments show that large full-attention Transformers continue to benefit from longer contexts, while efficient alternatives such as SSMs, linear attention variants, and sliding-window attention exhibit clearer saturation or degradation.

**Compliance With Llm Reviewing Policy:**

Affirmed.

**Key Questions For Authors:**

See weaknesses

**Limitations:**

yes

**Strengths And Weaknesses:**

# Strengths
The paper asks an important question and proposes a clean, well-motivated benchmark. The synthetic setup is interpretable, technically thoughtful, and useful for isolating architectural long-context capacity. The empirical comparison across multiple architecture families is also valuable.

# Weaknesses
1. Limited real-world validation.

Although L-CUBE is a clean synthetic benchmark, the paper does not sufficiently show that performance on this task predicts long-context performance on real language tasks. As a result, the practical significance of the benchmark is still somewhat unclear.

2. Architecture conclusions may be influenced by training setup.

While the benchmark is designed to isolate architectural capacity, some empirical differences may still be affected by optimization, scaling regime, or specific implementation choices, which makes purely architectural attribution somewhat less definitive.

3. Dependency structure is relatively narrow.

Although L-CUBE supports several information-scaling regimes, the benchmark still relies on a specific family of hierarchical Gaussian constructions, so it remains unclear how broadly the conclusions transfer to other synthetic or real long-range dependency structures.

4. The writing can be made more concise.

Some parts of the paper are overly segmented. For instance, the one-sentence paragraphs around Lines 382 and 426 seem unnecessary, and the conclusion section is divided into too many short paragraphs. Merging them would improve readability and overall flow.

---

> ### Author Rebuttal · Authors · 2026-03-31
>
> We thank the reviewer for recognizing the importance of the question we ask, the clean and well-motivated benchmark design, and the value of the empirical comparison across multiple architecture families.
>
> > **Weakness 1.**
>
> We thank the reviewer for raising this point. We wanted to clarify that performing well on L-CUBE is an important minimum requirement for an architecture to perform well on real long-context tasks. This is established information-theoretically, and this guarantee is already much stronger than what many existing benchmarks can provide. Common existing benchmarks such as needle-in-a-haystack, RULER, and long-document QA are motivated by human intuitions and are neither necessary nor sufficient conditions for real-world performance. In contrast, our benchmark provides a clean, information-theoretically grounded minimum that an architecture must satisfy in order to effectively model long-context data. As Reviewer D8w2 has also pointed out, our results are consistent with existing empirical findings. We will include additional discussion on the relation and differences between our benchmark and existing ones to better clarify this point.
>
> > **Weakness 2.**
>
> We thank the reviewer for this comment. Indeed, isolating architectural capacity from optimization and other hyperparameter effects is important, and this is actually well-supported by our synthetic dataset generator. Because we have access to the generator, we work in an effectively infinite-data regime and train all models for 500,000 iterations with the same optimization recipe, ensuring they converge well before training stops. In https://anonymous.4open.science/api/repo/lcube-rebuttal/file/rebuttal_figures.pdf?v=6ff58087, we include additional results on training convergence (Figures R5–R7), which show that all models converge well before the end of training. We also include results under multiple seeds (Figure R10) to demonstrate the robustness of the benchmark. These results will be included in the revision, along with additional clarifications on how our setup isolates optimization-related confounding factors.
>
> > **Weakness 3.**
>
> We thank the reviewer for this comment. We wanted to clarify that the main empirical focus on the sub-volume law regime is deliberate, as it is the regime most relevant to natural language as established by the L2M theory (Chen et al., 2025). It is the main target of the benchmark rather than simply one example among many. The additional regimes that the generator is capable of realizing demonstrate the generality of our construction, but they are not central to our paper's claim.
>
> With that said, we include new benchmark results with logarithmic scaling in https://anonymous.4open.science/api/repo/lcube-rebuttal/file/rebuttal_figures.pdf?v=6ff58087. In Figure R8, we show the utilization of representative vanilla attention, SSM, and log-linear attention models in this regime. Both vanilla attention and log-linear attention track the available gain closely, while the SSM fails at long sequences, consistent with theoretical predictions. In Figure R9, we plot the raw KL-divergence with similar trends. We will include more polished results in the revision, as some benchmarks in this regime were still running. However, we note that the comprehensive benchmark in the sub-volume law case already cost approximately 80,000 H100 GPU hours, and it is impractical to conduct equally comprehensive benchmarks in all additional settings. We will include logarithmic scaling results as demonstrative examples in the revision to show the generality of L-CUBE.
>
> > **Weakness 4.**
>
> We thank the reviewer for the specific suggestions. We will merge the short paragraphs around Lines 382 and 426 and consolidate the conclusion section as recommended to improve readability and flow.
>
>
> We thank the reviewer again for the constructive review. We hope our responses have addressed the concerns raised. We would appreciate it if the reviewer would consider raising the score or confidence level, and we are happy to address any additional questions during the discussion period.

---

> > ### Author Rebuttal · Reviewer_jC8j · 2026-04-01
> >
> > Thanks for the response, which has largely addressed my concerns.

---

> > > ### Author Response · Authors · 2026-04-02
> > >
> > > It is great to hear that the reviewers' concerns have been addressed. We would sincerely appreciate it if the reviewer could consider raising the score or confidence level.

---

### Official Review · Reviewer_H9eH · 2026-03-12

**Soundness:** 3
**Presentation:** 4
**Significance:** 3
**Originality:** 3
**Overall Recommendation:** 4
**Confidence:** 4

**Summary:**

This paper introduces L-CUBE, a synthetic benchmark designed to isolate a model's architectural capacity for long-context modeling from its semantic knowledge and training data biases. Using hierarchical Gaussian sequences with controllable mutual information scaling, L-CUBE provides exact ground-truth conditionals. This allows for evaluation using conditional KL divergence rather than standard perplexity, offering a clean metric called "Long-Context Utilization" (LCU). The authors evaluate various architectures, including Transformers, State Space Models (Mamba), and hybrid models. The results demonstrate that while Transformers scale well with context, pure SSMs struggle to utilize information beyond their fixed state size unless augmented with attention mechanisms.

**Compliance With Llm Reviewing Policy:**

Affirmed.

**Key Questions For Authors:**

1. How sensitive are the evaluation results to the specific tokenization scheme used for the continuous Gaussian values? Could a different quantization or tokenization approach artificially help or hinder certain architectures?

2. While L-CUBE is a necessary condition, have you observed any empirical correlation between a model's LCU score on this synthetic task and its actual performance on standard natural language long-context benchmarks (e.g., RULER or zero-shot passkey retrieval) after pretraining?

3. For the hybrid architectures (like Jamba or RecurrentGemma), does the L-CUBE benchmark reveal any actionable insights into the optimal ratio or placement of attention layers versus SSM layers?

**Limitations:**

The authors appropriately address limitations in Section I, explicitly acknowledging that L-CUBE measures architectural capacity for hierarchical dependencies but does not evaluate semantic understanding, reasoning, or in-context learning. They correctly frame the benchmark as a necessary, but not sufficient, condition for long-context capabilities.

**Strengths And Weaknesses:**

Soundness

Strengths: The theoretical grounding is excellent. Using exact KL divergence from a known generative process eliminates the confounding variables of vocabulary and real-world semantic knowledge. The metric, Long-Context Utilization (LCU), is strictly defined and rigorously applied across multiple architectural families.

Weaknesses: While the synthetic continuous data is cleanly defined, tokenizing Gaussian sequences to feed into discrete language models could introduce distribution artifacts that differ significantly from natural language tokenization. Additionally, as the authors acknowledge, success on L-CUBE is a necessary but not sufficient condition for real-world long-context reasoning.

Presentation

Strengths: The paper is clearly written. The mathematical formulation of the hierarchical generator and the transition to the evaluation metrics are logically laid out. The visualizations, particularly the utilization curves comparing Transformers and SSMs, effectively communicate the core findings.

Weaknesses: The connection between the specific parameters of the hierarchical Gaussian tree (e.g., branching factor, correlation strength) and the resulting mutual information decay could be explained with more intuitive, non-mathematical examples early in the text to assist broader audiences.

Significance

Strengths: Diagnosing architectural flaws before committing massive compute resources to pretraining is a major pain point in foundation model research. L-CUBE provides a highly practical, computationally cheap diagnostic tool for researchers designing new efficient sequence modeling architectures.

Weaknesses: Because the benchmark tests pure dependency capture on synthetic data, its predictive power for downstream semantic tasks (like needle-in-a-haystack or long-document QA) remains indirect.

Originality

Strengths: Synthetic tasks for sequence modeling exist (e.g., associative recall, synthetic copying), but creating a scalable, hierarchical framework with exact, analytically tractable conditional probabilities to compute true KL divergence is a rigorous contribution to architecture evaluation.

Weaknesses: The broader concept of using synthetic Markov or hierarchical processes to test sequence models has been explored in prior statistical learning literature. The novelty here is primarily in the execution and scaling of these concepts as a formal LLM diagnostic tool, rather than inventing a new theoretical paradigm.

---

> ### Author Rebuttal · Authors · 2026-03-31
>
> We thank the reviewer for the thorough review and for recognizing the theoretical grounding, the use of exact KL divergence from a known generative process, and the rigorous application of the Long-Context Utilization metric across multiple architectural families.
>
> > **Soundness Weaknesses:**
>
> Regarding tokenizing Gaussian sequences: we wanted to clarify that we do not use any tokenization or quantization scheme. As discussed in Appendix E, we adapt the models with custom continuous embedding layers and Gaussian output heads suited for the continuous random variables, preserving the core architectural mechanisms being evaluated (attention, recurrent state updates, positional encoding) while enabling direct comparison against ground-truth conditionals. We will improve the appendix to make this adaptation clearer and the reference more prominent in the main text.
>
> Regarding the necessary but not sufficient condition: this property is independent of our use of continuous variables, and is in fact a central feature of L-CUBE by design. The goal is to isolate long-range dependency from other aspects of natural language, so the ability to capture long-range dependency is by definition a necessary condition for modeling natural language. The fact that our benchmark provides a necessary condition exactly demonstrates that we achieved this isolation goal. We will clarify this framing in the revision.
>
> > **Presentation Weaknesses:**
>
> We agree that providing more intuitive, non-mathematical examples early in the text would help broader audiences. We will update the paper accordingly. We appreciate specific suggestions from the reviewer.
>
> > **Significance Weaknesses:**
>
> L-CUBE concerns a different and more information-theoretically oriented aspect of long-context modeling than downstream semantic tasks like needle-in-a-haystack or long-document QA. All existing benchmarks are also proxies to real downstream usage, and it is unknown whether those benchmarks correspond directly to downstream tasks either (they are neither sufficient nor necessary conditions). Our benchmark is at least a necessary condition, has a clean information-theoretic interpretation, and concretely guides the design of next-generation efficient architectures. We will include more discussion on these relations in the revision.
>
> > **Originality Weaknesses:**
>
> Existing literature has explored synthetic Markov or hierarchical processes, but none satisfies both: (1) exact and efficient sampling and inference, and (2) reproducing the bipartite mutual information scaling of natural language. Markov constructions miss long-range dependencies by definition, and existing hierarchical processes (e.g., Lin & Tegmark, 2017) only address two-point mutual information, which the L2M paper (Chen et al., 2025) showed is insufficient to quantify long-range dependence structure. Systems with identical two-point scaling can have drastically different bipartite mutual information scaling, which governs long-context modeling requirements. Our construction provides both exact tractability and the correct bipartite scaling. We will include additional discussion in the revision.
>
> > **Q1.**
>
> As noted above, we do not use any tokenization scheme. We adapt models with continuous embedding layers and Gaussian output heads, which best helps isolate long-range dependency from everything else.
>
> If one were to tokenize the continuous variables, sufficiently dense quantization would likely preserve the mutual information scaling. However, one would lose efficient exact sampling and inference, an important property of our generator. We will include this discussion in the revision.
>
> > **Q2.**
>
> Our results are generally consistent with existing empirical findings, as Reviewer D8w2 has also noted. Our benchmark focuses on the information-theoretic perspective of long-context capacity, and correlation with downstream task performance is expected given that capturing long-range dependencies is a prerequisite for any such task. We will include additional discussion on these relations in the revision.
>
> > **Q3.**
>
> Hybrid architectures containing full-attention layers should behave similarly to vanilla attention on our benchmark, as even very small vanilla attention models capture the growing mutual information well. Asymptotically, hybrid models cannot avoid quadratic complexity. The goal of this paper is not to find the optimal ratio between global attention and other layers, but to test whether efficient alternatives (such as SSM or log-linear attention) can represent the power-law growth of mutual information, and to guide design of such alternatives. We will include a discussion on hybrid models in the revision.
>
> We thank the reviewer for the detailed and constructive review. We hope our responses have sufficiently addressed the concerns. We would be grateful if the reviewer would consider raising the score, and we are happy to address any further questions during the discussion period.

---

### Official Review · Reviewer_kWHZ · 2026-03-13

**Soundness:** 3
**Presentation:** 4
**Significance:** 3
**Originality:** 3
**Overall Recommendation:** 5
**Confidence:** 4

**Summary:**

This paper introduces **L-CUBE**, a synthetic benchmark for evaluating long-context language modeling through a controlled hierarchical Gaussian data generator. The key idea is to construct sequences with analytically tractable long-range dependencies, allowing the benchmark to vary how bipartite mutual information scales with context length and to compute exact conditional distributions. Using this setup, the authors propose a utilization-based evaluation framework that measures how much of the available contextual information different architectures actually extract. Experiments across Transformers, windowed attention models, SSMs, and linear-attention variants show clear differences in long-context utilization, suggesting that L-CUBE can serve as a diagnostic tool for isolating architectural limits in long-range modeling.

**Compliance With Llm Reviewing Policy:**

Affirmed.

**Final Justification:**

My concerns have been addressed.

**Key Questions For Authors:**

1. Can the authors provide model-side results across multiple information-scaling regimes, beyond the default sub-volume setting?
2. Can the authors provide a token-matched or FLOP-matched ablation for the utilization comparison between the full-context and no-context models?
3. Can the authors include results for hybrid attention architectures that combine sliding-window attention with global attention, and possibly also sparse attention variants? *(I understand that this may be difficult to add within the rebuttal period, which is completely understandable, but such results would strengthen the paper.)*

**Limitations:**

yes

**Strengths And Weaknesses:**

### Soundness

**Strengths.**

- The paper is technically coherent and well motivated. The benchmark construction, the utilization-based metric, and the goal of isolating architectural long-context capacity fit together cleanly.
- A key strength is analytical tractability: because the data distribution is controlled, the paper can evaluate exact conditional quantities rather than relying only on task accuracy or perplexity.
- The empirical results are broadly consistent with the intended diagnostic use case: full-attention Transformers, windowed models, SSMs, and linear-attention variants exhibit clearly different long-context utilization behaviors.

**Weaknesses.**

- The main empirical conclusions are demonstrated mostly under the default sub-volume scaling regime. Although the generator can realize multiple regimes, the paper does not yet show enough model-side results across them.
- The utilization comparison is based on training full-context and no-context models on different sequence lengths with otherwise similar recipes. This leaves some room for a training-budget confound, since part of the gain may come from token/compute differences rather than contextual access alone.
- The empirical section would also be stronger with more robustness evidence, such as seed variance or convergence behavior,

### Presentation

The paper is clearly written and easy to follow. The motivation, benchmark design, metric, and experiments are presented in a logical order.

### Significance

**Strengths.**

- The paper addresses an important problem. As long-context modeling becomes increasingly central, controlled benchmarks that isolate architectural limits are valuable. I can see this being useful for researchers as a diagnostic tool before investing in much larger-scale downstream experiments.

**Weaknesses.**

- Its practical significance depends on whether performance on L-CUBE correlates with real long-context behavior on natural data, and that connection is not yet established strongly enough.

### Originality

The paper is meaningfully original in its combination of a controllable hierarchical generator, explicit information-scaling regimes, and a utilization-based evaluation framework.

---

> ### Author Rebuttal · Authors · 2026-03-31
>
> We thank the reviewer for the thorough and detailed review, and for acknowledging the coherence and motivation, the analytical tractability, and the diagnostic value of our empirical results.
>
> > ### Strengths And Weaknesses
> > **Soundness Weakness 1**
>
> The reviewer raised a very good point. We wanted to clarify that the focus on the sub-volume law regime is deliberate, as it is the regime most relevant to natural language and the main target of the benchmark rather than simply a default example. The additional regimes demonstrate generality but are not central to our claim.
>
> With that said, we include new results with logarithmic scaling in https://anonymous.4open.science/api/repo/lcube-rebuttal/file/rebuttal_figures.pdf?v=6ff58087. In Figure R8, we show utilization of representative vanilla attention, SSM, and log-linear attention models. Theoretically, both vanilla attention and log-linear attention can represent logarithmically growing mutual information, while SSMs are limited to constant mutual information. This is exactly shown in the results: both vanilla attention and log-linear attention track the available gain closely, while the SSM fails at long sequences. Figure R9 shows raw KL-divergence with similar trends.
>
> We will include more polished results in the revision, as some benchmarks in this regime were still running. However, we note that the comprehensive sub-volume law benchmark already cost around 80,000 H100 GPU hours, so we are only able to include additional demonstrative examples in the logarithmic regime to show the generality of L-CUBE.
>
> > **Soundness Weakness 2**
>
> The reviewer raised a sharp point. In our setup, the generator provides effectively unlimited fresh samples, so experiments are much closer to an infinite-data regime. All models are trained with the same optimization recipe for 500,000 iterations. In https://anonymous.4open.science/api/repo/lcube-rebuttal/file/rebuttal_figures.pdf?v=6ff58087, we include evaluation results at earlier training stages (Figures R1-R4) and training curves (Figures R5-R7), both demonstrating that models converge well before the end of training. Working in the regime where all models have converged isolates data- and optimization-related confounding factors and makes result interpretation much cleaner. We will further clarify this in the revision.
>
> > **Soundness Weakness 3**
>
> We agree that more robustness evidence would strengthen the results. However, repeating each experiment many times is impractical given the resources required. To demonstrate robustness, we show in Figure R10 of https://anonymous.4open.science/api/repo/lcube-rebuttal/file/rebuttal_figures.pdf?v=6ff58087 representative training curves under different seeds. The results remain largely the same under different seeding conditions. The training curve convergence in Figures R5-R7 further supports robustness. These results will be included in the revision.
>
> > **Significance Weakness 1**
>
> We appreciate this important point. Our benchmark identifies the minimum requirements an architecture must satisfy to perform well on long-context natural data, and therefore has a direct connection to real long-context behavior. While a full-scale empirical correlation study is an important direction, it is somewhat outside the scope of the current work. As Reviewer D8w2 has also pointed out, our results are consistent with existing empirical findings. We will further clarify this in the revision.
>
> >### Key Questions For Authors
> >**1.**
>
> We addressed this in our response to soundness weakness 1 above. Please see that response for our new logarithmic-scaling results (Figures R8 and R9 in [link]).
>
> >**2.**
>
> We addressed this in our response to soundness weakness 2. Because we work in the infinite-data regime and train all models to convergence (Figures R5-R7, R1-R4), confounding effects of token or compute budget differences are effectively eliminated.
>
> >**3.**
>
> Hybrid architectures are indeed practical alternatives to full attention models. However, the main question in this paper is whether a given mechanism can carry the required growing mutual information. Hybrid architectures containing full-attention layers should behave similarly to the vanilla attention family, as even very small vanilla attention models capture the growing mutual information well. Asymptotically, hybrid models also cannot avoid quadratic complexity. The focus of this paper is more on whether an efficient alternative (such as SSM or log-linear attention) can represent the power-law growth of mutual information, and on using this benchmark to guide the design of such alternatives. We will include a discussion on hybrid models in the revision.
>
> We thank the reviewer for the thoughtful review. We hope our responses have sufficiently addressed the concerns. We would be very grateful if the reviewer would consider raising the score, and we are happy to address any further questions during the discussion period.

---

> > ### Author Rebuttal · Reviewer_kWHZ · 2026-04-01
> >
> > My concerns have been addressed.

---

> > > ### Author Response · Authors · 2026-04-02
> > >
> > > It is great to hear that we have resolved the concerns and thanks for raising the score!

---

### Official Review · Reviewer_D8w2 · 2026-03-17

**Soundness:** 4
**Presentation:** 3
**Significance:** 3
**Originality:** 3
**Overall Recommendation:** 5
**Confidence:** 4

**Summary:**

This paper serves as an empirical validation of the L2M theory of long-context language modeling, which predicts when an architecture should fail in long-context settings based on its state capacity, including a formalized relationship between bipartite mutual information and history state dimensions. A core contribution of the work is isolating long-range dependency modeling from world knowledge, a conflation that commonly appears in standard long-context benchmarks. To achieve this, they construct a synthetic hierarchical dataset that emulates the structure of natural data while offering controllable mutual information scaling, allowing them to precisely control the depth of long-range dependencies. Using this benchmark (L-CUBE), they run experiments across a variety of architectures including transformers, state-space models, and efficient alternatives, showing that the L2M capacity theory aligns well with their empirical findings. Importantly, they argue that L-CUBE can surface architectural limitations and inform design decisions even before training on real data, and notably, some of these limitations are revealed empirically rather than being derived purely from theory.

**Compliance With Llm Reviewing Policy:**

Affirmed.

**Key Questions For Authors:**

Questions:
1. Have the authors analyzed how the context utilization curves evolve during pre-training across the different architectures? It would be interesting to know whether certain architectures acquire long-range dependency capabilities earlier in training than others, and whether that correlates with the L2M capacity predictions. Any insight here would strengthen the paper's practical implications for architectural design choices.
2. Regarding lines 368-374, when applying rotary positional embeddings in the NeoX setup, did the authors use vanilla RoPE or also apply extrapolation methods such as YaRN? Since RoPE on its own does not extrapolate beyond its training context length.
3. Is there a possibility of extending L-CUBE evaluations to full bidirectional attention mechanisms? This would be particularly relevant for multimodal architectures and embedding models where full attention over all tokens is standard.

**Limitations:**

yes

**Strengths And Weaknesses:**

Summary of Strengths:

S1. The paper is well written and does a fine job outlining the methodology clearly. Despite being fairly heavy on information theory and math, they cover the necessary background sufficiently enough that the reader doesn't need to constantly chase down prior literature to follow along.

S2. The algorithm design for constructing the controllable scaling benchmark is intuitive and well motivated, particularly the expansion and reuse mechanisms that drive the hierarchical structure.

S3. The motivation is also understandable. L-CUBE is framed as a sufficiency test: if an architecture fails here, it is highly likely to fail on standard real long-context benchmarks as well. This is a clean and useful way to reveal architectural limitations.

S4. The extra analyses covering model scaling, dimensionality scaling, and sequence length are thorough, and I particularly appreciated the compute-context length utilization relationship conclusion they draw from the results---even though I have a small comment on that.

Summary of Weaknesses:

W1. The most critical gap in this paper is the missing discussion of prior literature on association recall tasks and knowledge-isolated synthetic benchmarks. Several works have approached long-range dependency evaluation through fully synthetic, knowledge-free setups and reached overlapping conclusions [1,2]. For instance, [1] arrives at a similar finding both theoretically and empirically on association recall tasks, including similar to the conclusion that a 124M GPT-2 model can utilize longer contexts than a 1.4B Mamba model. While these works are not grounded in L2M theory, they cover highly relevant architectural failure modes and their absence from the related work significantly hinders the perceived novelty of this paper.

[1] Arora, Simran, Sabri Eyuboglu, Aman Timalsina, Isys Johnson, Michael Poli, James Zou, Atri Rudra, and Christopher Ré. "ZOOLOGY: MEASURING AND IMPROVING RECALL IN EFFICIENT LANGUAGE MODELS."

[2] Poli, Michael, Stefano Massaroli, Eric Nguyen, Daniel Y. Fu, Tri Dao, Stephen Baccus, Yoshua Bengio, Stefano Ermon, and Christopher Ré. "Hyena hierarchy: Towards larger convolutional language models." In International Conference on Machine Learning, pp. 28043-28078. PMLR, 2023.

W2. A more fundamental concern is whether long-context dependencies and knowledge limitations are truly fully separable. It is plausible that knowledge limitations, while absent in short-context cases, could show up specifically in long-context settings [3]. If that is the case, the core premise of L-CUBE as a clean isolated test is harder to defend, and this deserves at least a discussion. The examples of coding, DNA processing, etc... all still require some-level of syntactical/semantical understanding.

[3] Modarressi, Ali, Hanieh Deilamsalehy, Franck Dernoncourt, Trung Bui, Ryan A. Rossi, Seunghyun Yoon, and Hinrich Schuetze. "NoLiMa: Long-Context Evaluation Beyond Literal Matching." In Forty-second International Conference on Machine Learning.

W3. Finally, lines 432-434 in the conclusion ground a suggestion on preliminary results that are only presented in the appendix. It is generally discouraged to anchor conclusions on appendix content that falls outside the main scope of the paper. If these results are strong enough to support a conclusion, they should be incorporated into the body of the paper, even as a minor result, with the extended details referred to the appendix, similar to the approach taken in line 426 and its prior paragraph.

---

> ### Author Rebuttal · Authors · 2026-03-31
>
> We thank the reviewer for the thorough review of our manuscript and for appreciating the clarity of our paper, the benchmark construction, and the usefulness of L-CUBE as a controlled sufficiency test for long-context capacity.
> > ### Summary of Weaknesses
> >
> > **W1.**
>
> We thank the reviewer for pointing out the missing discussion of association recall tasks and other knowledge-isolated synthetic benchmarks. We agree these papers are relevant and should be discussed in the related works. We will include them in the revision. At the same time, we wanted to clarify that L-CUBE is looking at a different object. The main point of this work is not to simply reproduce the long-known empirical fact that transformers tend to perform better than SSMs on long-context modeling. Instead, the point here is to provide a benchmark with exact ground-truth conditionals, controlled bipartite mutual information scaling matched to natural language, and a direct information-theoretic interpretation based on the L2M theory. This gives us the ability to not only study whether an architecture succeeds or fails at long context, but also understand its failure mode in terms of the scaling of its effective history state size. In addition, we note that L-CUBE is not restricted to the specific setting that reproduces the power-law scaling behavior observed in natural language, but can realize a wide range of mutual information scaling regimes. We will revise the related works section to include more association recall papers, including the papers mentioned by the reviewer, and we will also include additional discussion on the connections and differences between them and our work.
> > **W2.**
>
> We agree with the reviewer that this point should be phrased more carefully. Our claim is not that long-context dependency and knowledge are fully independent. Instead, we are making the point that the architectural ability to capture long-range dependence is a distinct minimum requirement, which should be isolated from other confounding factors including knowledge for clean testing. In other words, some knowledge in real tasks may indeed live at long range, but before a model can use such knowledge, it still must be able to represent the relevant long-range dependencies at all. L-CUBE is designed to isolate this prerequisite. We will revise the wording in the paper to make it more precise and explicit.
> > **W3.**
>
> We agree with the reviewer on this point. We will incorporate the results into the main text in our revision.
>
> >### Key Questions For Authors
> > **1.**
>
> The reviewer makes a very good suggestion. In https://anonymous.4open.science/api/repo/lcube-rebuttal/file/rebuttal_figures.pdf?v=6ff58087, we include additional results at intermediate stages of training (Figures R1-R4). While during earlier stages of training all model utilizations are lower, the general behavior of different models stays largely similar to the results at the end of the training. In addition, the gap at earlier stages is larger for SSMs, which is consistent with L2M capacity predictions. We also show the evaluation loss during training in Figures R5-R7. It can be seen that all models converge well before the end of the training. We will include these results in the revision of the manuscript.
>
> > **2.**
>
> We wanted to first clarify that GPT-NeoX uses the vanilla RoPE setting and does not apply extrapolation methods such as YaRN. However, the key point here is not about whether extrapolation methods are used, but rather that even when an architecture family satisfies the asymptotic L2M condition based on a direct counting of the history state dimension, specific design choices such as positional encoding can still affect the empirical long-context utilization and create an "effective" history state size that differs from the theoretical one. This is also one reason why L-CUBE is useful in addition to the existing theory, which cannot diagnose such implementation-level effects. We will include these clarifications in the revision of the manuscript.
>
> > **3.**
>
> Yes, the generator itself does not have a preferred direction, and the Gaussian conditional statistics can in principle be efficiently evaluated in other conditional orders as well. The current paper focuses on autoregressive evaluation as it most directly relates to the L2M theory, but the benchmark itself is not restricted to this specific use case. We will add a short discussion noting that L-CUBE can also support bidirectional or alternative masking settings in the next revision.
>
> We thank the reviewer for the detailed review. We hope these clarifications adequately address the reviewer's concerns. We would truly appreciate it if the reviewer would consider raising the score. We are happy to answer any further questions during the discussion period.

---

> > ### Author Rebuttal · Reviewer_D8w2 · 2026-04-01
> >
> > A_Q1: Thanks for adding these results. Very interesting, especially the evolution of the utilization curves of SSM. Did windowed attention get worse (w=64)?
> > One more note: also maybe for the final manuscript add more finer steps in the beginning.
> >
> > Looking forward to the final version of your paper and I will adjust my scores accordingly.

---

> > > ### Author Response · Authors · 2026-04-01
> > >
> > > Thanks a lot for raising the score!
> > >
> > > Regarding windowed attention, we would like to clarify that the utilization is only a proxy for how much mutual information the model captures. When the value is close to the available gain, this proxy is well behaved; however, when the value is far from the available gain, the interpretation becomes less straightforward, as it may no longer monotonically correlated with the captured mutual information. With that said, in the case of windowed attention at w=64, we believe what have happened is that early in training, the model has learned very little and captures only a small, noisy amount of mutual information. As training progresses, the window size limitation forces the model to prioritize local patterns over global structure, which can adversely affect utilization, which specifically measures whether the model captures long-range dependency.
> > >
> > > We will note this explicitly and include the discussion in the revision. We will also add finer-grained checkpoints in the early stages of training.

---

### Decision · Program_Chairs · 2026-04-30

**Decision:**

Accept (regular)

**Comment:**

This paper introduces L-CUBE, a synthetic benchmark for evaluating the long-context capabilities of LLLMs independently of semantic knowledge and vocabulary statistics. L-CUBE disentangles dependency-capturing capacity from semantic knowledge by leveraging hierarchical Gaussian sequences with controllable bipartite mutual information scaling. The authors further propose a new metric called long-context utilization to quantify how effectively models extract predictive information with increasing  context length. Experiments on  multiple architectures validate the theoretical predictions and suggest that L-CUBE can provide a practical tool for assessing long-context performance before expensive  training. Reviewers highlighted the strong motivation of the paper, its technical coherence and theoretical grounding, sound  algorithms design, and the value of empirical comparison across different architectures.  There were also several concerns-- such as omission of some prior work, practical significance of the findings on real-world tasks and  transferability to a broader set of long-range dependency structures-- which were addressed during the rebuttal. Overall, reviewers agree that this work represents a substantial and technically rigorous contribution and deserves acceptance.